# Systemically administered wound-homing peptide accelerates wound healing by modulating syndecan-4 function

Horacio Maldonado [1], Bryan D. Savage[1], Harlan R. Barker [2], Ulrike May[2], Maria Vähätupa[2], Rahul K. Badiani[1], Katarzyna I. Wolanska [1], Craig M. J. Turner[1], Toini Pemmari[2], Tuomo Ketomäki[2], Stuart Prince[2], Martin J. Humphries [3], Erkki Ruoslahti[4], Mark R. Morgan [1,5] ✉ & Tero A. H. Järvinen [2,4,5] ✉

CAR (CARSKNKDC) is a wound-homing peptide that recognises angiogenic neovessels. Here we discover that systemically administered CAR peptide has inherent ability to promote wound healing: wounds close and re-epithelialise faster in CAR-treated male mice. CAR promotes keratinocyte migration in vitro. The heparan sulfate proteoglycan syndecan-4 regulates cell migration and is crucial for wound healing. We report that syndecan-4 expression is restricted to epidermis and blood vessels in mice skin wounds. Syndecan-4 regulates binding and internalisation of CAR peptide and CAR-mediated cytoskeletal remodelling. CAR induces syndecan-4-dependent activation of the small GTPase ARF6, via the guanine nucleotide exchange factor cytohesin-2, and promotes syndecan-4-, ARF6- and Cytohesin-2-mediated keratinocyte migration. Finally, we show that genetic ablation of syndecan-4 in male mice eliminates CAR-induced wound re-epithelialisation following systemic administration. We propose that CAR peptide activates syndecan-4 functions to selectively promote re-epithelialisation. Thus, CAR peptide provides a therapeutic approach to enhance wound healing in mice; systemic, yet target organ- and cell-specific.

Large numbers of protein-based therapeutics that could potentially enhance tissue regeneration have been identified, such as growth factors, but their therapeutic value in clinical medicine has been limited due to the difficulty of maintaining bioactivity of locally applied proteins in the protease-rich environment of regenerating tissues[1]. Although human diseases are often treated with systemically administered drugs, the systemic use of powerful biological agents has largely been ruled out due to safety concerns. Thus, pharmaceutical efforts aimed at enhancing tissue repair have been based on local application at the site of the injury[2].

To facilitate systemic delivery in regenerative medicine, we previously sought to create a platform for systemic administration and target-specific delivery that is based on wound-homing peptides. The peptides were identified through screening of phage libraries in vivo for peptides that home to angiogenic blood vessels in skin wounds and transected tendon injuries[3]. A 9-amino acid cyclic peptide, CAR (sequence CARSKNKDC) was particularly effective in homing to wounds and shown to recognise angiogenic blood vessels in regenerating tissues[3]. Subsequently, CAR peptide was used to deliver a recombinant fusion protein, consisting of the anti-fibrotic protein

[1]Institute of Systems, Molecular & Integrative Biology, University of Liverpool, Liverpool, UK. [2]Faculty of Medicine and Health Technology, Tampere University & Tampere University Hospital, Tampere, Finland. [3]Wellcome Trust Centre for Cell-Matrix Research, University of Manchester, Manchester, UK. [4]Cancer Center, Sanford Burnham Prebys Medical Discovery Institute, La Jolla, CA and Center for Nanomedicine, University of California (UCSB), Santa Barbara, CA, USA. [5]These authors jointly supervised this work: Mark R. Morgan, Tero A. H. Järvinen. ✉e-mail: mark.morgan@liverpool.ac.uk; tero.jarvinen@tuni.fi

decorin, into healing wounds[4]. CAR-mediated targeting enhanced the accumulation and anti-fibrotic activity of decorin in the wounds; promoting wound healing and suppressing scar formation[4]. Along with CAR peptide, other peptides, protein fragments and antibodies capable of homing to regenerating tissues have been characterised for regenerative medicine applications[5]. These homing vehicles have been successfully used for enhanced delivery of therapeutic recombinant proteins, nanoparticles loaded with drugs, extracellular vesicles or even stem cells to injured tissue[5]. The most clinically advanced homing peptides/antibodies are in phase II and III clinical trials in oncology[6].

Here, we have investigated the effect of CAR peptide on wound healing and show that CAR alone, in the absence of a coupled therapeutic partner, has an inherent ability to promote wound healing in male mice. This observation led us to investigate how CAR accelerates wound repair. We found that CAR has little effect on granulation tissue formation but promotes re-epithelialisation of wounds and epithelial cell migration selectively in male mice. We previously reported that CAR peptide requires heparan sulfate proteoglycans (HSPGs) for cell binding and penetrating activity[3,4]. Syndecan-4 (SDC4) is a wound repair-promoting transmembrane HSPG, which functions as an extracellular matrix (ECM) receptor and growth factor co-receptor[7–9]. SDC4 co-ordinates spatiotemporal regulation of small GTPase activity, in response to the extracellular microenvironment, to modulate cell migration[7–13]. In this study we identify SDC4 as a key regulator of CAR-dependent wound healing in male mice. Mechanistically, SDC4 is required for CAR peptide internalisation, CAR-mediated cytoskeletal reorganisation, ARF6 activity modulation, keratinocyte migration and wound repair. These results suggest that CAR peptide accelerates wound healing in mice selectively by activating endogenous SDC4 promigratory mechanisms in epithelia.

## Results

### CAR peptide accelerates skin wound re-epithelialisation and closure

Skin wound closure occurs when keratinocytes migrate from the edge of the wound and re-epithelialise the epidermis[14,15]. CAR peptide is capable of targeting angiogenic vasculature in injured and inflamed tissues enabling delivery to sites of injury[3,16–18]. When CAR-decorin wound healing studies were conducted[4], preliminary indications of accelerated wound re-epithelialisation in the animals treated with CAR peptide alone (a control group to treatment with recombinant CAR-decorin fusion protein) were noted, even at a very low dose of CAR peptide ($1 \mu g/kg$ daily)[4]. Therefore, we examined the potential therapeutic effect of systemic CAR peptide administration on skin wound healing. Systemic intravenous (i.v.) administration of CAR or mutant CAR peptides (mCAR; CA**RSK**NKDC mutated to CA**QSN**NKDC in mCAR, which abolishes the homing activity almost completely)[3,4], or BSA in PBS (Control) via tail vein injections was initiated 24 h post-wounding. The treatment was continued for 5, 7 or 10 days in three independent treatment experiments with 24 mice in each treatment group. Based on previous CAR peptide-studies[16–18], a treatment regimen of two daily doses of 3.0 mg/kg CAR peptide was selected.

Daily analysis of the wounds demonstrated that wounds treated with CAR peptide closed significantly faster than the control groups during the treatment trials (Fig. 1a, b). The wounds were significantly smaller in CAR-treated animals than in control groups from day 4 onwards ($P < 0.05$ CAR vs control; $P < 0.001$ CAR vs mCAR), exhibiting substantial decreases in open wound area from day 7 to 10 (Fig. 1a, b). Moreover, the percentage of wounds that showed complete closure was significantly higher in CAR peptide-treated animals than in control groups (Fig. 1c, d).

Having established a clear effect on macroscopic wound closure, histological analyses were carried out to determine the impact of CAR peptide on wound re-epithelialisation, contraction and granulation tissue formation. Excision wounds close via co-ordination of re-epithelialisation and wound contraction (approximately 60% and 40% contribution, respectively, in BALB/c mice) and re-epithelialisation and contraction can be reliably quantified[19] (Fig. 2a). Histological analysis of wounds on days 5, 7 and 10 showed that re-epithelialisation was significantly more advanced in the group that received CAR peptide than in the control groups. The size of the gap between epidermal tongues ($WG_x$), i.e., the region of wound remaining without epidermis, was 60% and 83% smaller in CAR-treated than in control groups 5 and 7 days after wounding, respectively (Fig. 2b, h). The length and the area of hyperproliferative epidermal (HPE) tongues were assessed at the earliest time point (day 5). The length of HPE tongues ($E1_x + E2_x$), as well as the HPE area, were significantly higher in CAR peptide-treated wounds than in control or mCAR-treated mice (Fig. 2c, d). However, the total width of the wounds ($W_x$) was similar in different treatment groups (Fig. 2e). Next, the proportion of wounds with complete re-epithelialisation was determined. The percentage of wounds that had complete re-epithelialisation was significantly higher in the CAR-treated group than in control groups at all time-points examined (Fig. 2c). All CAR peptide-treated wounds had completed re-epithelialisation by day 10, whereas part of the wounds remained without new epidermis in control and mCAR-treated wounds (Fig. 2b, f, h).

After re-epithelialisation has taken place, wounds contract in size. While the widths of the wounds ($W_x$) were identical in all treatment groups on days 5 and 7, they were significantly smaller in CAR peptide-treated mice on day 10 (Fig. 2e). Together these results demonstrate that CAR-induced wound closure and re-epithelialisation were not caused by panniculus carnosus-driven wound contraction and demonstrate accelerated maturation of the wound following early completion of re-epithelialisation in CAR-treated wounds. While granulation tissue looked more mature in CAR peptide-treated animals at early time-points (days 5 and 7), as suggested by reduced fibrin clot, no difference in granulation tissue area was detected between the groups at any studied time-point (Fig. 2g, h). Again, these data are consistent with CAR peptide not inducing wound contraction. Thus, CAR peptide accelerates wound healing and maturation primarily by selectively promoting re-epithelialisation.

### CAR peptide promotes epithelial cell migration

To elucidate the mechanism by which CAR peptide accelerates wound re-epithelialisation, we performed immunohistochemistry (IHC) analyses of skin wounds treated with systemic i.v. injections of CAR, mCAR and BSA/PBS (Control) and collected the tissue on days 5, 7 and 10 post-wounding (Figs. S1; S2). Proliferation within the whole activated epidermis, including both the proliferation zone and the newly formed migrating epidermis was assessed. We detected no significant differences in the numbers of Ki67-positive proliferating cells in epidermis or granulation tissue between the treatment groups at any studied time-point during the healing period (Fig. S1a, b, f). Accumulation of macrophages in granulation tissue was also similar between the treatment groups (Fig. S1c, f). Next, we analysed wound vascularisation (angiogenesis): a slightly higher density of blood vessels was detected in early granulation tissue in CAR peptide-treated wounds on day 5, but this was minor and did not persist at later timepoints (Fig. S1d, f). To test the effect of CAR on myofibroblast transformation and subsequent wound contraction, we determined the number of cells expressing high levels of α-smooth muscle actin (α-SMA), i.e., myofibroblasts, in the granulation tissue (Fig. S1e, f). Interestingly, CAR peptide-treated wounds had substantially fewer myofibroblasts than other treatment groups; indicating that enhanced wound closure in the CAR-treated group was not due to myofibroblast-driven contraction. Imaging α-SMA staining in day 7 wounds, demonstrated that no myofibroblasts were present in granulation tissue in any of the treatment groups (Fig. S3). Further indicating that the increased wound repair in CAR-treated mice was not a result of premature myofibroblast activation.

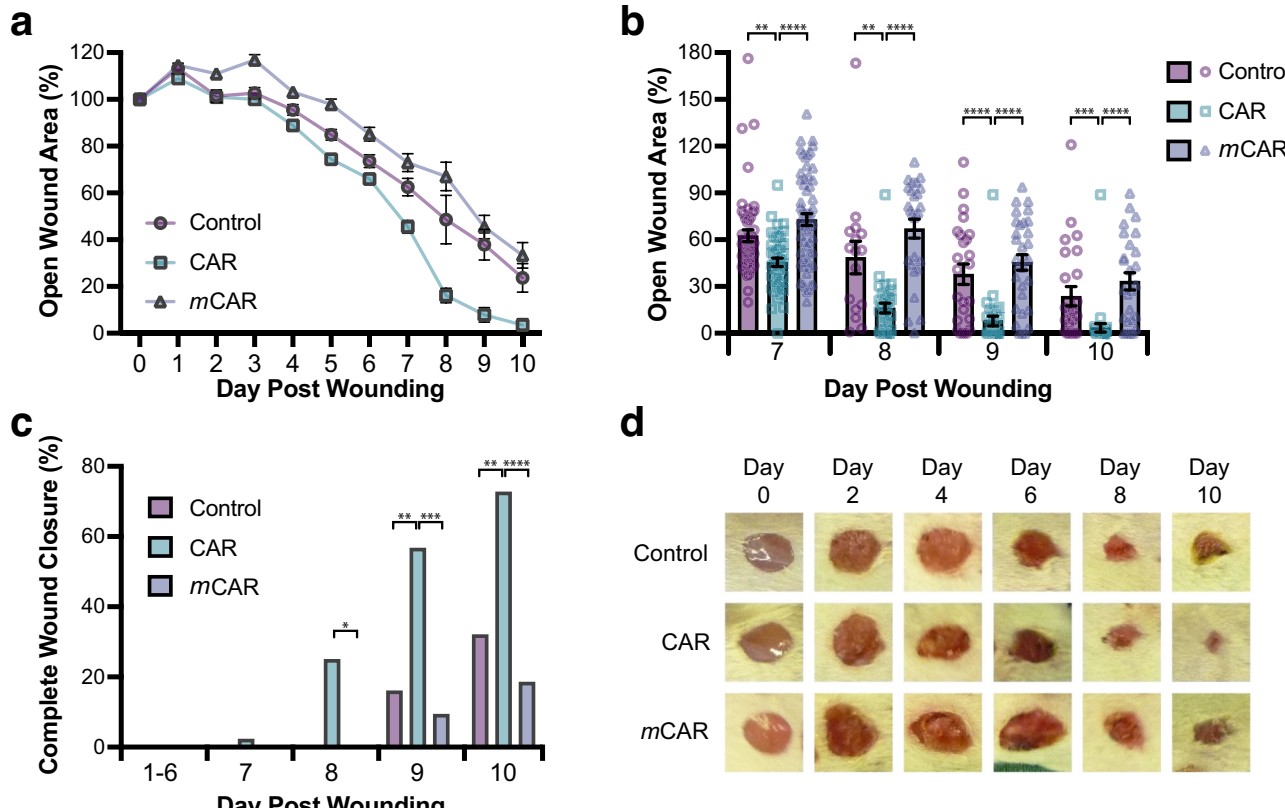

**Fig. 1 | CAR peptide accelerates wound closure.** Mice with full thickness skin excision wounds were treated i.v. twice a day with CAR, mCAR or control (BSA/PBS) injections from day one post-wounding until the sacrifice as described in methods. Wounds were examined and photographed daily. Wound closure was recorded and expressed (**a, b**) as the percentage of the open wound size relative to its original size on the day of wounding (Day 7: CAR vs. Control $P = 0.0041$, CAR vs. mCAR $P = 2.0 \times 10^{-06}$; Day 8: CAR vs. Control $P = 0.009$, CAR vs. mCAR $P = 9.5 \times 10^{-08}$; Day 9: CAR vs. Control $P = 0.00027$, CAR vs. mCAR $P = 1.3 \times 10^{-06}$; Day 10: CAR vs. Control $P = 0.001$, CAR vs. mCAR $P = 1.6 \times 10^{-06}$) or (**c**) as the percentage of the number of completely closed wounds (Day 8: CAR vs. Control $P = 0.078$ (not significant), CAR vs. mCAR $P = 0.017$; Day 9: CAR vs. Control $P = 0.0049$, CAR vs. mCAR $P = 0.0003$; Day 10: CAR vs. Control $P = 0.0064$, CAR vs. mCAR $P = 0.0001$). **d** Representative macroscopic digital pictures of the wounds treated with CAR, mCAR and control peptide injections are shown at different time-points of the wound closure process. There were 24 animals, each with four wounds, in every treatment group. Source data are provided as a Source Data file. Values are mean ± S.E.M. Each data point represents an individual wound. $n = 96$ (Days 0–5), 60 (Days 6–7) and 32 (Days 8–10) wounds in each treatment group. *$P \leq 0.05$; **$P \leq 0.01$; ***$P \leq 0.001$; ****$P \leq 0.0001$. Krustal-Wallis rank sum test with Dunn´s test with tie correction (**a, b**) and Pearson´s Chi-square test (without continuity correction) with post-hoc test Fisher exact test (two-sided) (**c**).

Thus, CAR peptide enhances re-epithelialisation but does not enhance cell proliferation, macrophage accumulation or vascularisation in wounded skin tissue. Moreover, increased activation of myofibroblasts was not observed in CAR-treated wounds. These data suggest that CAR peptide enhances wound healing by direct and selective stimulation of the wound epidermis. These results led us to hypothesise that CAR-mediated wound healing is driven by promotion of epidermal cell migration. Live-cell imaging of keratinocyte migration, using HaCaT cells in scratch wound assays, revealed that CAR peptide accelerated keratinocyte migration on fibronectin. Over a 20-h period, CAR peptide enhanced scratch wound closure approximately two-fold, relative to mCAR peptide or vehicle controls (Fig. 3a; Supplementary Movie 1). The ability of cells to migrate across a substrate is influenced by both the speed and directionality of migration[20,21]. Cell tracking analysis revealed that CAR peptide significantly increased the speed of cell migration (Fig. 3b) but had no effect on directional persistence (Fig. 3c). Interestingly, when we assessed cell migratory profiles, CAR-stimulated cells often exhibited an initial suppression of cell motility relative to controls, followed by a significant surge in migration while mCAR and vehicle treated controls slowed down (Fig. 3d, e; Supplementary Video 2). Together, these data suggest that CAR peptide accelerates wound healing in vivo by promoting keratinocyte migration.

## SDC4 regulates CAR uptake and cytoskeletal re-organisation

We have previously shown that CAR peptide requires HSPGs for cell binding and penetrating activity[3,4]. As the transmembrane receptor SDC4 is a HSPG that regulates cell migration and wound healing, we sought to determine whether SDC4 might be involved in the cellular and wound-promoting effects of CAR peptide. We initially used a well-characterised mouse embryonic fibroblast (MEF) model that has previously been used to dissect SDC4-dependent mechanisms: comprised of immortalised wild-type MEFs (Im$^{+/+}$), SDC4$^{-/-}$ MEFs (Syn4−/−) and SDC4 re-expressing (Syn4−/− expressing near endogenous levels of human syndecan-4) MEFs (Syn4WT)[20]. Incubation with fluorescently labelled CAR-FAM (10 μg/ml) gave substantially higher levels of cell-associated fluorescence in Im$^{+/+}$ and Syn4WT MEFs, compared with Syn4−/− MEFs (Fig. 4a). Importantly, in Syn4WT cells, CAR peptide internalised into intracellular vesicular structures, where it colocalised with SDC4 (Fig. 4a, b, bi).

Cell migration relies critically on the co-ordination of cytoskeletal and adhesion dynamics. So, we next analysed the effect of CAR peptide on cytoskeletal and adhesion complex organisation. Cells were plated on the central cell-binding domain of fibronectin (50 K/Fn6-10), prior to stimulation with CAR, mCAR or vehicle control. This approach, which prevents matrix-engagement of SDC4, enables experimental separation and interrogation of

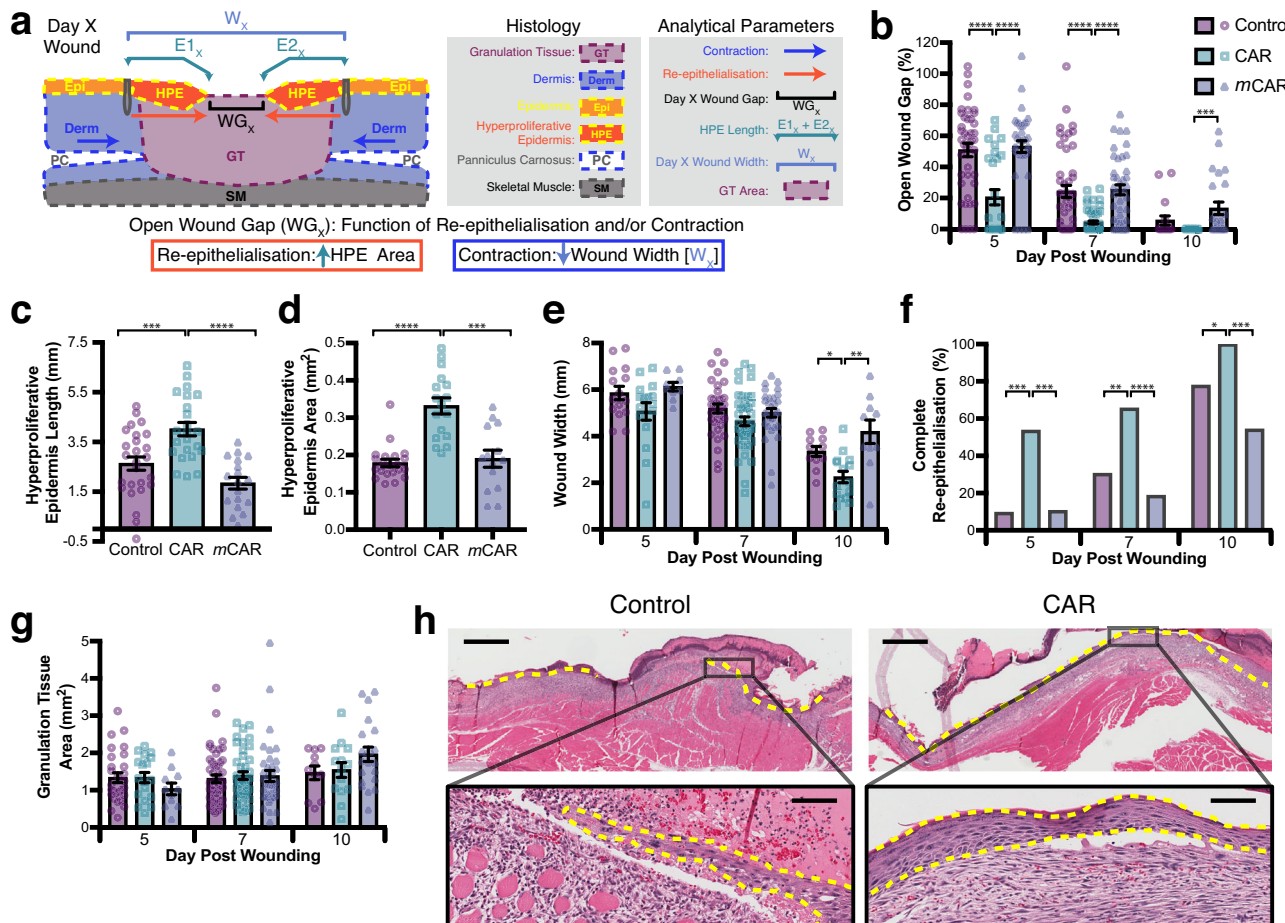

**Fig. 2 | CAR peptide stimulates wound re-epithelialisation.** Mice with full thickness skin excision wounds were treated with systemically administered CAR, *m*CAR and control (BSA/PBS) peptide as described in methods. Wounds were harvested on days 5, 7 and 10. **a** Schematic representation of histological analysis. Open wound gap: WG$_X$ - Gap between epithelial tongues on day X post-wounding. Re-epithelialisation: E1$_X$ + E2$_X$ - Hyperproliferative epidermis (HPE) length. Contraction: Wound width reduction W$_X$. **b** Gap between the epidermal tongues (WG$_X$: area of open wound without re-epithelialisation) represented as percentage (%) of the original wound size (Day 5: CAR vs. Control $P = 6.3 \times 10^{-05}$, CAR vs. *m*CAR $P = 6.3 \times 10^{-07}$; Day 7: CAR vs. Control $P = 1.3 \times 10^{-05}$, CAR vs. *m*CAR $P = 1.3 \times 10^{-07}$; Day 10: CAR vs. *m*CAR $P = 0.00025$). **c** Length of hyperproliferative epidermal tongues at day 5 (Day 5: CAR vs. Control $P = 0.005$, CAR vs. *m*CAR $P = 1.1 \times 10^{-05}$). **d** Area of hyperproliferative epidermal tongues at day 5 (CAR vs. Control $P = 3.8 \times 10^{-06}$, CAR vs. *m*CAR $P = 0.00029$). **e** Overall wound width (W$_X$), indicating wound contraction (Day 10: CAR vs. Control $P = 0.028$, CAR vs. *m*CAR $P = 0.0023$).

**f** Percentage of completely re-epithelialised wounds (Day 5: CAR vs. Control $P = 0.0002$, CAR vs. *m*CAR $P = 0.0005$; Day 7: CAR vs. Control $P = 0.0013$, CAR vs. *m*CAR $P = 1.0^{-07}$; Day 10: CAR vs. Control $P = 0.041$; CAR vs. *m*CAR $P = 0.0002$). **g** Cross-sectional area of granulation tissue quantified by examining two histological HE-stained sections from each wound. **h** Representative histological HE-stained pictures of the wounds treated with CAR peptide and control collected on day 7 illustrate wound re-epithelialisation. The epidermal tongues are marked with yellow dashed lines. Scale bars: 300 μm low magnification image, 30 μm high magnification inset. **a**−**h** Eight animals, each with four wounds, in every treatment group (N = 28, 44, 24 for days 5, 7 and 10). Source data are provided as a Source Data file. Each data point represents an individual wound. Values are mean ± S.E.M. Kruskal-Wallis rank sum test with Dunn´s test with tie correction (**b**, **c**, **d**, **e** and **g**) and Pearson´s Chi-square test (without continuity correction) with post-hoc test Fisher exact test (two-sided) (**f**).

integrin- and syndecan-dependent signals and functions, as described previously[11,13,20].

Treatment of cells expressing endogenous or re-expressed wild-type SDC4 induced a distinctive cytoskeletal reorganisation within 30 min of stimulation with CAR, characterised by loss of actin stress fibres, formation of actin-rich membrane ruffles, dissolution/disassembly of α5β1-dependent adhesion complexes and accumulation of α5β1 in intracellular vesicles (Fig. S4a, bi). However, while a subset of cells continued to exhibit this morphology, following 120 min of CAR stimulation approximately 80% of SDC4-expressing cells underwent further cytoskeletal reorganisation and exhibited pronounced stress fibres and elevated levels of α5β1 integrin at the cell-matrix interface (Fig. S4a, bi). By contrast, *m*CAR treatment had no discernible effect on cytoskeletal architecture or integrin distribution, compared with the vehicle control. Importantly, Syn4−/− cells did not undergo

cytoskeletal or adhesion complex reorganisation following CAR peptide treatment (Fig. S4bi, bii). Thus, stimulation of cells with CAR peptide induces cytoskeletal and α5β1 integrin adhesion complex remodelling in a SDC4-dependent manner.

The observation that CAR regulates SDC4-dependent adhesion complex remodelling, led us to investigate whether CAR initiates similar remodelling events in keratinocytes engaging full-length fibronectin. Quantitative analyses demonstrated that 5 min of CAR treatment triggered a rapid loss of α5β1 integrin from well-established adhesion complexes. However, 30 min of sustained CAR treatment, was sufficient to enable redelivery of α5β1 integrin to the cell-matrix interface in robust adhesion complexes (Fig. 4c, d). Together, these data suggest that CAR peptide regulates adhesion dynamics, cytoskeletal remodelling and the ability of α5β1 integrin to modulate interactions with fibronectin.

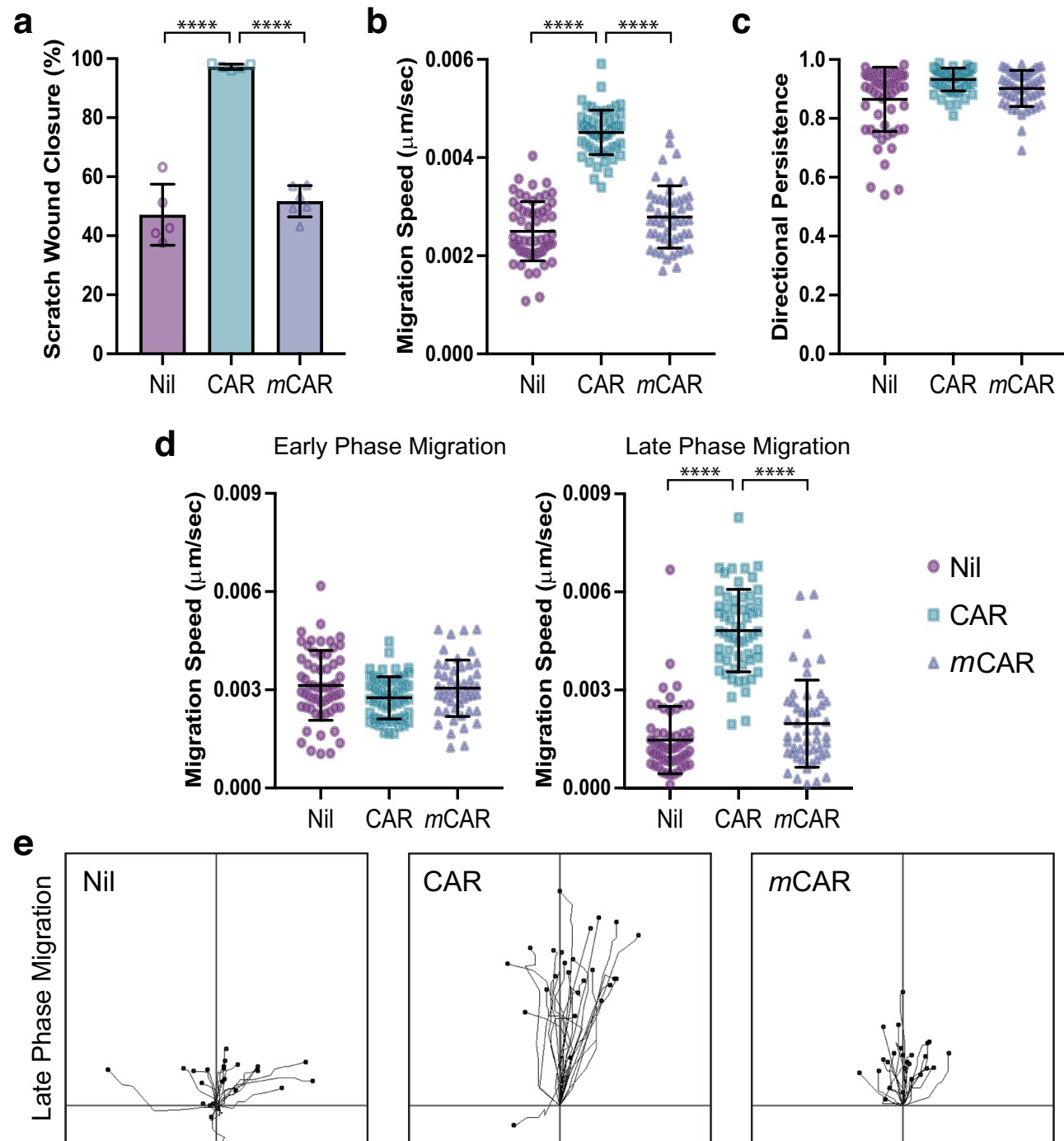

**Fig. 3 | CAR peptide stimulates keratinocyte migration.** Migration of HaCaT keratinocytes on fibronectin in scratch wound assays, in the presence or absence of 10 µg/ml CAR or mCAR peptide. Cells were analysed over 20 h by time-lapse microscopy. **a** Scratch wound closure, **b** mean migration speed throughout time-lapse, **c** directional persistence throughout timelapse, **d** speed at early and late phases of migration (Early: Timepoint 0–5 h; Late: Timepoint 15–20 h), and **e** representative migration tracks during late phase migration. Data are representative from one of three independent experiments. Values are means ± S.D. All statistical analyses are two-way ANOVA with Tukey's multiple comparisons test.

**a** $n = 5$–6 fields of view per condition (Nil $n = 5$, CAR $n = 5$, mCAR $n = 6$); Nil vs CAR $P = 6.133 \times 10^{-8}$; CAR vs mCAR $P = 1.147 \times 10^{-7}$. **b–d** $n = 50$–60 cells per condition (Nil $n = 60$, CAR $n = 59$, mCAR $n = 50$). **b** Nil vs CAR $P = 1.5 \times 10^{-14}$; CAR vs mCAR $P = 1.5 \times 10^{-14}$. **d** Early phase: Nil vs CAR $P = 0.0499$; CAR vs mCAR $P = 0.1932$. Late phase: Nil vs CAR $P = 1.5 \times 10^{-14}$; CAR vs mCAR $P = 1.7 \times 10^{-14}$. **a** Each data point represents a single field of view; **b–d** Each data point represents an individual cell. Source data are provided as a Source Data file. See also Supplementary Movies S1 & S2.

## SDC4 and fibronectin are selectively expressed in migrating epidermis of skin wounds

As analyses in well-characterised cell models of SDC4-mediated functions suggested that SDC4 is required for CAR peptide internalisation, and that CAR regulates SDC4-dependent cytoskeletal and adhesion dynamics, we assessed SDC4 expression in wounds. Fibronectin is the main endogenous SDC4 ECM ligand that activates SDC4-dependent pro-migratory pathways[7,11,13] and is expressed abundantly in skin wounds. Thus, we explored the expression and distribution of SDC4 and fibronectin in skin wounds. In line with previous studies[22],

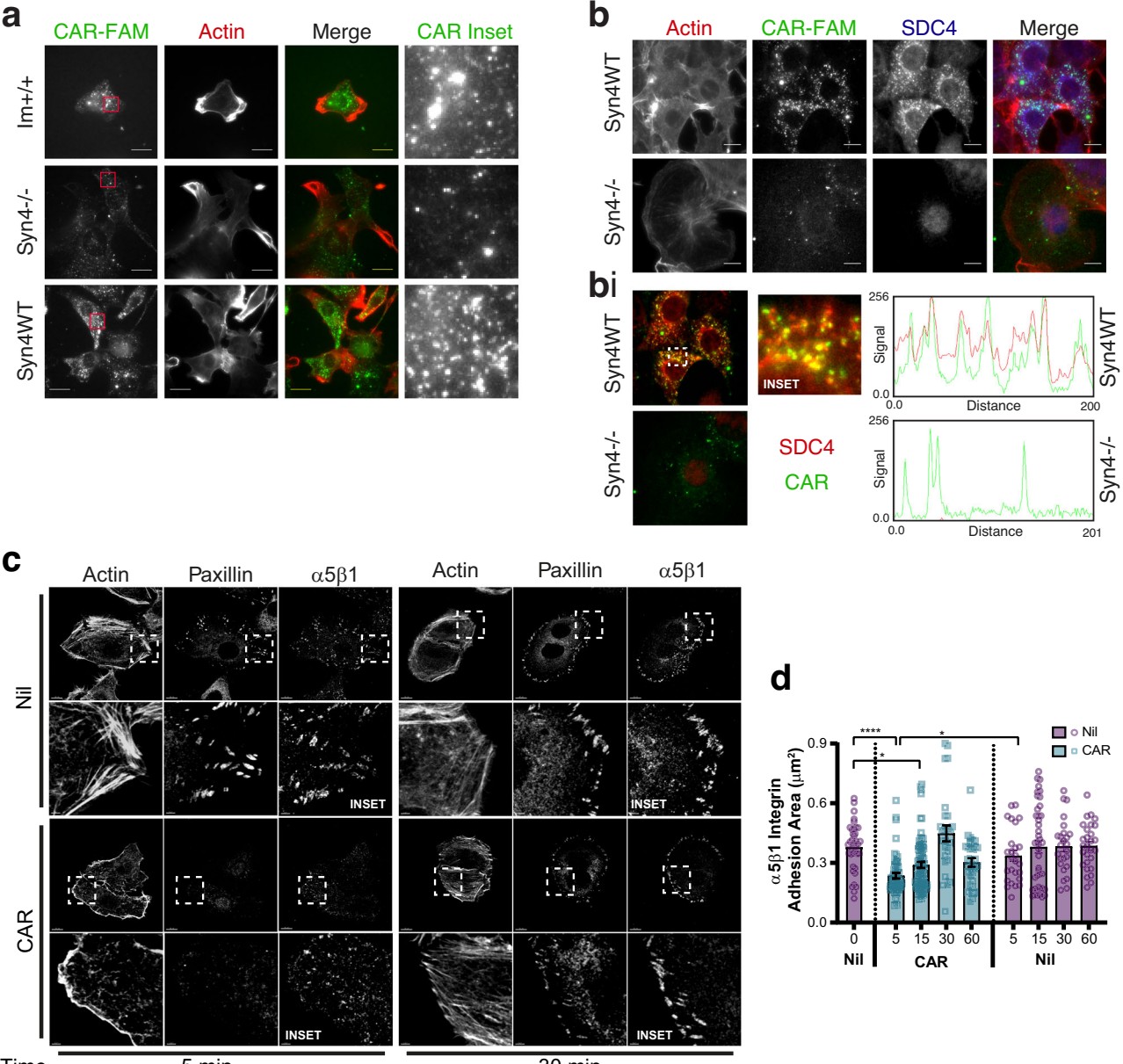

**Fig. 4 | CAR peptide is internalised by SDC4 and modulates α5β1 integrin adhesion complex dynamics. a** Subcellular distribution of fluorescently-labelled CAR in immortalised wild-type MEFs (Im⁺), syndecan-4-/- MEFs (Syn4-/-) and syndecan-4 re-expressing MEFs (Syn4WT) following 8 h treatment. CAR-FAM (Green) and actin (phalloidin-AlexaFluor-594; Red). Maximum projections of 2.1 μm z-sections are displayed. Scale bar = 20 μm. *N* = 2 independent replicate experiments. **b** Immunofluorescence demonstrating that internalised CAR co-localises with SDC4. CAR-FAM internalisation in Syn4WT and Syn4-/- MEFs. Cells were treated with CAR peptide for 8 h prior to fixation and stained for SDC4 and actin. CAR (Green), SDC4 (Blue) and actin (phalloidin-AlexaFluor-594; Red). Sum projections of 2.1 μm z-sections are displayed. Scale bar = 10 μm. *N* = 2 independent replicate experiments. **bi** Co-localisation of SDC4 and CAR in intracellular vesicles. Images correspond to Fig. 5b; sum projections of 0.6 μm z-section, positioned 0.6–1.2 μm above cell-matrix interface (central region of cells) pseudo-coloured to highlight co-localisation CAR-FAM (Green), SDC4 (Red). Dashed box highlights inset region. RGB profiles: fluorescence intensity of CAR and SDC4 along 25.8 μm segmented line intersecting CAR-FAM positive vesicles. **c, d** Quantitative analysis of α5β1 integrin and paxillin-positive adhesion complexes in fibronectin-bound keratinocytes following CAR peptide stimulation (*N* = 3 independent replicate experiments). **c** Subcellular distribution of α5β1 integrin, paxillin and actin in HaCaT cells on fibronectin following 5- or 30-min treatment with 10 μg/ml CAR or vehicle control. Dashed boxes indicate inset regions (depicted in lower image). Scale bars: 10 μm (main images); 2 μm (inset images). **d** Area of α5β1-positive integrin adhesions (μm²/cell) ± S.E.M. following 0-, 5-, 15-, 30- and 60-min treatment with 10 μg/ml CAR or vehicle control. Data points represent mean α5β1 integrin-positive adhesion area per cell. *N* = 3 with 26–75 images analysed per condition (Control: 0 min *n* = 38, 5 min *n* = 26, 15 min *n* = 43, 30 min *n* = 25, 60 min *n* = 28; CAR: 5 min *n* = 53, 15 min *n* = 75, 30 min *n* = 27, 60 min *n* = 31). Kruskal-Wallis test, followed by Dunn's multiple comparisons test: Nil 0' vs CAR 5' *P* = 7.555 × 10⁻⁶; Nil 0' vs CAR 15' *P* = 0.0113; CAR 5' vs Nil 5' *P* = 0.0447. Source data are provided as a Source Data file.

antibody staining revealed strong SDC4 expression in migrating epidermis (epidermal tongues), some expression in the dermis, mainly in blood vessels, but also in the granulation tissue (Fig. 5a, f). IHC double staining for SDC4 and fibronectin showed that, where SDC4 was

localised within migrating epidermis, the expression pattern of fibronectin is the opposite: exhibiting no expression in migrating epidermis but abundant expression throughout underlying granulation tissue (Fig. 5b, c, ci). Thus, while SDC4 was expressed throughout the

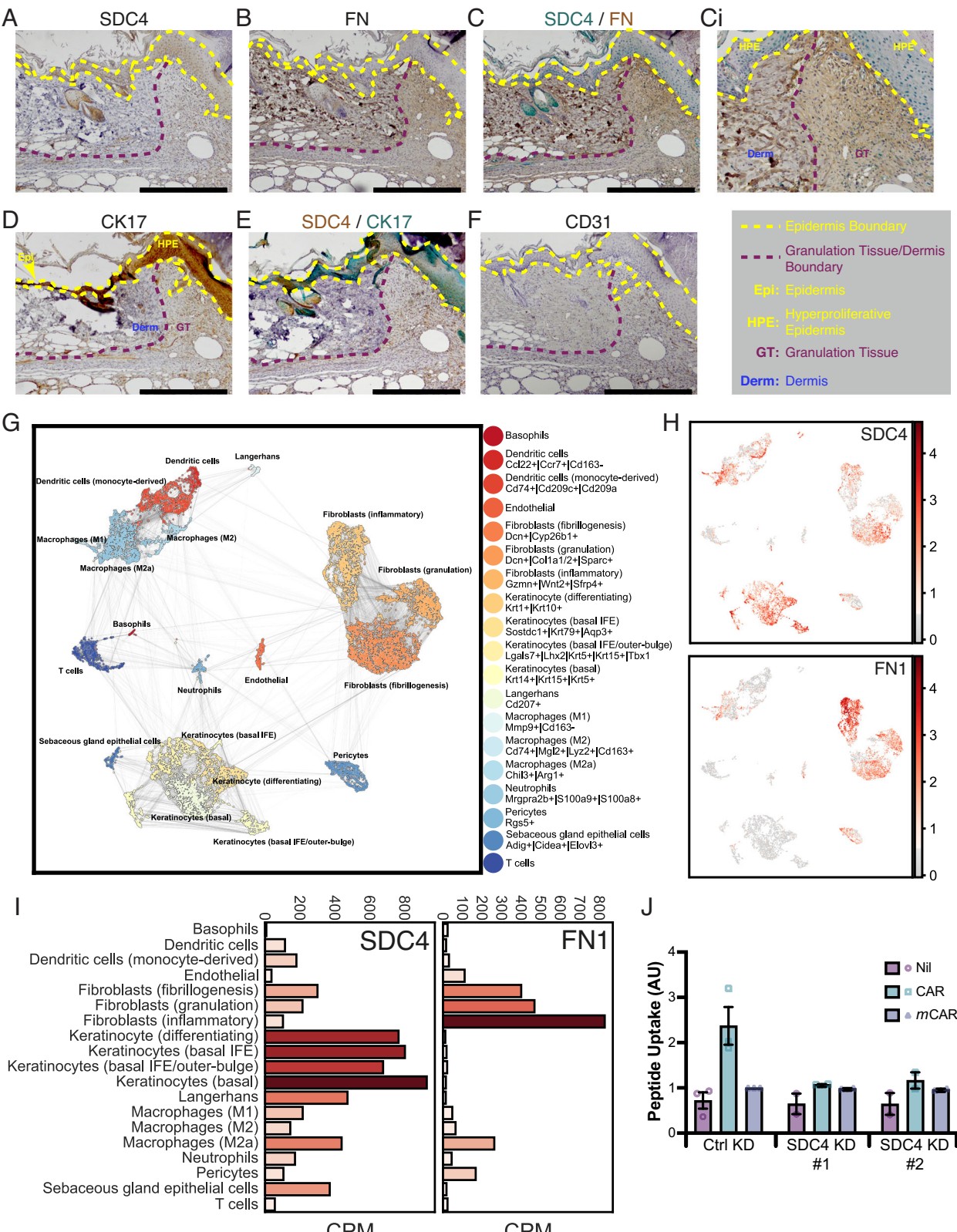

epidermal tongues, only cells directly contacting granulation tissue (basal keratinocytes) would have an opportunity to interact with fibronectin. Double staining of SDC4 and keratinocyte-specific marker cytokeratin 17 (CK17), confirmed the epithelial distribution of SDC4 and demonstrated co-expression in the migrating epidermis of skin wounds (Fig. 5d, e).

To further understand how CAR peptide might modulate wound healing at a cellular level, we analysed single-cell transcriptomics data published on wound healing, in order to delineate the specific cell types within wounds that express high levels of key regulatory molecules. We selected data from ref. 23 for interrogation because the study reports single-cell RNA sequencing (scRNA-Seq) data from

**Fig. 5 | SDC4 is highly expressed in wound keratinocytes and required for CAR peptide uptake.** IHC analysis of full thickness skin excision wounds. Samples were harvested from untreated mice 7 days post wounding and representative micrographs are presented of skin wound sections stained for (**a**) SDC4, (**b**) fibronectin (FN), (**c**) and (**ci**) SDC4 (Green) and fibronectin (Brown), (**d**) cytokeratin-17 (CK17), (**e**) SDC4 (Brown) and cytokeratin-17 (CK17) (green), (**f**) blood vessels (CD31). **ci** High magnification image of a region within the same field of view as (**c**). Epidermis is highlighted with yellow dashed lines. Boundary between granulation tissue and dermis is marked by red dashed lines. HPE: Hyperproliferative epidermis; Epi: Epidermis; GT: Granulation tissue; Derm: Dermis. SDC4 signal intensity is not directly comparable between (**a**) and (**e**). **a**–**f** $n = 12$ individual wounds; three animals with four wounds. Scale bar: 500 µm. **g**–**i** Single-cell RNA-Seq analysis of SDC4 and fibronectin expression in skin wounds. Single-cell transcriptomics analysis of gene expression in different cells in skin wounds[23]. scRNA-Seq data from 16,351 cells

was analysed using the SCANPY Python library[72] and clusters identified using Louvain clustering at resolution 0.75. **g** Clusters were visualised with the UMAP model and cell types determined by literature review of highly upregulated genes in each Louvain cluster. **h** Syndecan-4 (SDC4) and Fibronectin (FN1) expression mapped onto all cells. **i** Barplots presenting average counts per million (CPM) across each identified cell type are given for both genes. **j** Uptake of fluorescent-labelled peptides by HaCaT cells transfected with control siRNA (CTRL KD), human SDC4-targeting siRNA oligo #1 (SDC4 KD #1) or human SDC4-targeting siRNA oligo #2 (SDC4 KD #2), treated with CAR-FAM, $m$CAR-FAM or vehicle control (Nil) for 4 h. Data are from 2 to 3 independent replicate experiments (CTRL KD: $N = 3$; SDC4 KD #1 & #2: $N = 2$) and normalised relative to the $m$CAR-FAM signal in CTRL KD cells. Each data point represents mean uptake in each independent experiment ± S.D. Source data are provided as a Source Data file.

normal and wounded mouse skin for all major skin cell populations (Fig. 5g)[23]. Our analysis of the scRNA-Seq skin wound dataset revealed very high SDC4 expression in all keratinocyte populations involved in wound healing, in contrast to substantially lower expression in different fibroblast populations (Fig. 5h, i). By contrast, expression of fibronectin (FN1) is largely restricted to fibroblasts, especially those in an inflammatory state (Fig. 5h, i). Together, the IHC and scRNA-Seq analyses suggest that SDC4 is primarily expressed in the migrating epidermis in healing mouse wounds, whereas fibronectin synthesis is restricted to underlying fibroblast populations.

As SDC4 is selectively expressed in wound epithelia, CAR promotes keratinocyte migration and CAR can use SDC4 to enter cells, flow cytometry was used to quantitatively determine the role of SDC4 in CAR peptide uptake in keratinocytes. Uptake of CAR-FAM peptide by HaCaT cells was significantly higher than $m$CAR-FAM and siRNA-mediated suppression of SDC4, with two different oligonucleotide sequences, reduced CAR-FAM uptake to basal levels (Fig. 5j). Thus, SDC4 is required for the uptake and internalisation of CAR peptide in epithelial cells.

## CAR peptide regulates SDC4-dependent ARF6 activity
The vesicular distribution of CAR and SDC4 (Fig. 4b, bi), suggested that their functional relationship may involve trafficking mechanisms. SDC4 is a direct regulator of receptor recycling and ligand-engagement of SDC4 modulates activation of the small GTPase ARF6 to spatially and temporally control integrin recycling and matrix-engagement[13,24]. ARF6 is activated upon ECM engagement and regulates recycling of receptors from intracellular vesicles to the plasma membrane[13,25], and we have shown that SDC4 controls ARF6 activity to regulate differential recycling of integrins to co-ordinate cell migration[13,24].

The fact that SDC4 is required for internalisation of CAR (Figs. 4a, b; 5J), co-localises with internalised CAR peptide (Fig. 4b) and regulates the rapid CAR-dependent redistribution of α5β1-integrin adhesion complexes (Fig. 4c, d; S5), and that CAR promotes epithelial cell migration (Fig. 3), prompted us to test whether CAR peptide regulates SDC4-dependent ARF6 activity in keratinocytes. Effector pull-down assays in HaCaT cells showed that stimulation with CAR peptide dynamically regulated ARF6 activity; inducing initial suppression of ARF6 activation, followed by a subsequent activation of ARF6 that was approximately 2.5-fold higher than in $m$CAR-treated cells (Fig. 6a, b). Likewise, CAR stimulation in HaCaT cells transfected with non-targeting control siRNA rapidly suppressed ARF6 activity by approximately 25%, followed by a substantial increase in ARF6 activity that was significantly higher than baseline and control conditions. However, siRNA-mediated knockdown of SDC4 abrogated the ability of CAR to regulate ARF6 activity (Fig. 6c–e). Thus, these time-course experiments demonstrate that CAR peptide dynamically controls ARF6 activation, resulting in ARF6 hyperactivity, and that SDC4 is essential for this effect.

Having demonstrated a link between CAR peptide and SDC4-mediated regulation of ARF6 activity, we analysed ARF6 expression in the scRNA-Seq skin wound healing dataset[23]. Interestingly, the highest levels of ARF6 expression were detected in differentiating and basal keratinocytes, (Fig. S5a, b); two cell types with a key role in wound re-epithelialisation[26] and high levels of SDC4 expression (Fig. 5h, j). Demonstrating that keratinocytes in mouse excisional skin wounds express high levels of the key molecules associated with the CAR-stimulated signalling response.

We next assessed the impact of depleting ARF6 expression on CAR-mediated cytoskeletal and integrin-mediated adhesion reorganisation. Intriguingly, under steady-state conditions on the integrin-binding central cell-binding domain of fibronectin (50 K/Fn6-10) and in the absence of peptide stimulation, siRNA-mediated depletion of ARF6 resulted in loss of actin stress fibres, formation of actin-rich membrane ruffles, and small and/or indistinct α5β1-dependent adhesion complexes (Fig. S6a, b). These morphological changes are reminiscent of the CAR-induced cytoskeletal and integrin reorganisation elicited by short-term stimulation with CAR peptide (Fig. S4a, b); a treatment that initially suppresses ARF6 activity (Fig. 6). Moreover, ARF6 knockdown inhibited the stress fibre and α5β1-dependent adhesion complex formation triggered by long-term CAR stimulation (Fig. S6a, b). Given that CAR stimulation induces an initial suppression of SDC4-dependent ARF6 activity followed by a substantial SDC4-dependent increase in ARF6 activity, these data suggest that the effect of CAR peptide on cytoskeletal and adhesion dynamics is dependent on both ARF6 and SDC4.

## CAR promotes SDC4- and ARF6-dependent keratinocyte migration
The preceding data demonstrated that CAR peptide, which promotes wound healing (Figs. 2, 3), is internalised with SDC4, induces SDC4-dependent ARF6 activation and triggers SDC4- and ARF6-dependent cytoskeletal reorganisation (Figs. 4, 6, S4, S6). As in vivo and cell biological data suggested that CAR induces wound healing by increasing epidermal cell migration (Figs. 2, 3), we sought to determine whether SDC4 and ARF6 are required for CAR-dependent epidermal cell migration.

To analyse the contribution of SDC4 to CAR-stimulated epithelial cell migration, we assessed the effect of siRNA-mediated suppression of SDC4 expression on HaCaT motility on fibronectin in scratch wound migration assays, following treatment with CAR, $m$CAR or vehicle control. SDC4 depletion, using two independent siRNA oligonucleotides, completely eliminated the increase in scratch wound closure and epithelial cell migration triggered by CAR peptide stimulation (Fig. 7a–d, S7A, Supplementary Movies 3 and 4). As observed previously, in control cells CAR modulated migration speed, but had no effect on directionality. Likewise, SDC4 knockdown also did not affect directional persistence (Fig. S7b). As we previously observed in wild-type cells, in control siRNA cells (Ctrl KD) CAR triggered an initial

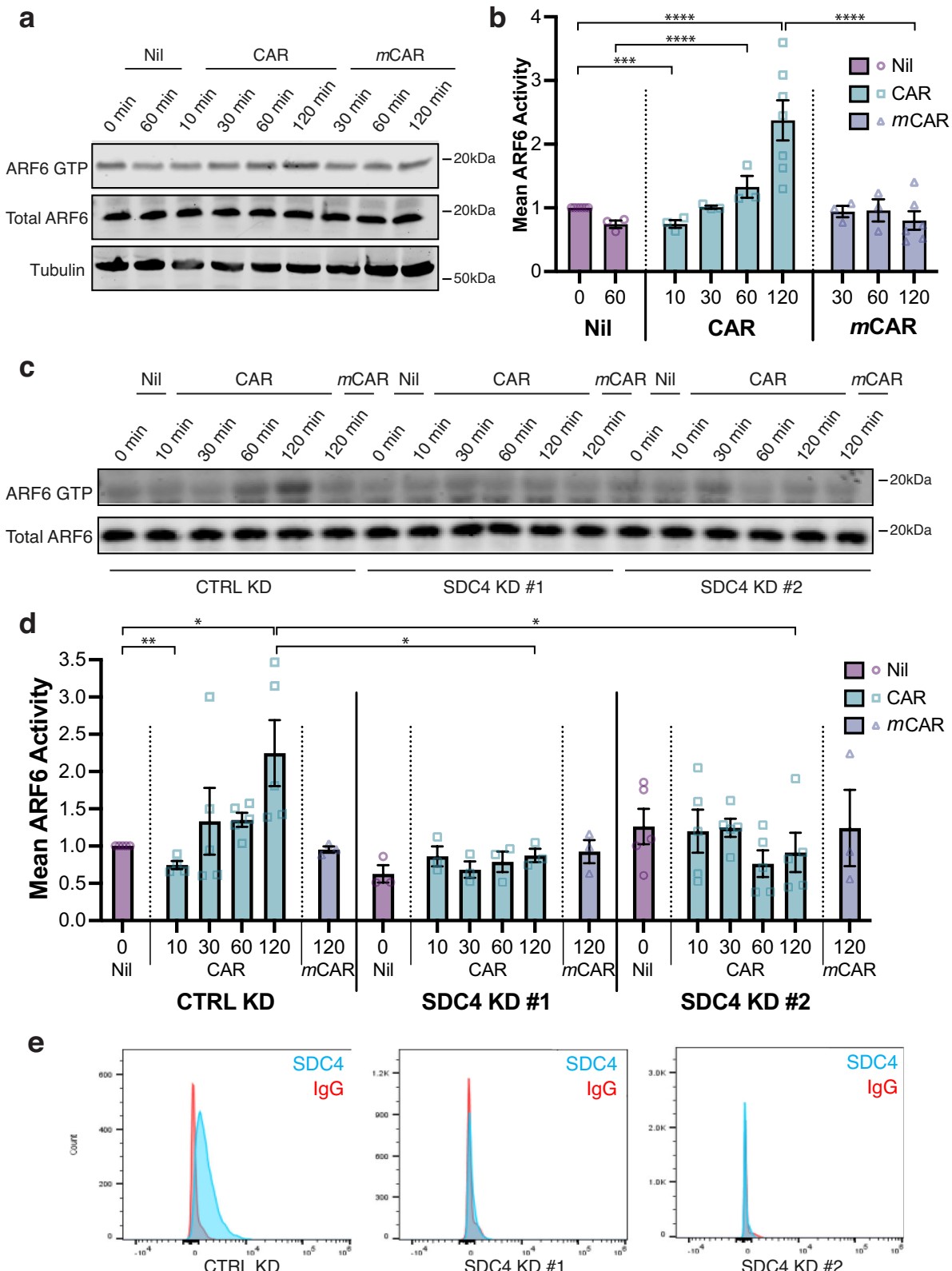

inhibition of migration followed by a surge in cell motility. Interestingly, both the initial CAR-induced suppression and subsequent increase in migration were dependent on SDC4 expression (Fig. 7b–d; Supplementary Movies 3 and 4).

As SDC4 is required for CAR-stimulated ARF6 activation and keratinocytes express high levels of ARF6 in wounds (Fig. S5), we next assessed cell migration following ARF6 knockdown. These experiments revealed that, while CAR peptide accelerates scratch wound closure and epithelial cell migration in non-targeting siRNA-expressing cells, depletion of ARF6 inhibited CAR-dependent migration (Fig. 7e–h; S7c, d; Supplementary Movies 5 and 6). Thus, SDC4 regulates CAR-mediated ARF6 activity and both SDC4 and ARF6 are required for the pro-migratory effects of CAR peptide in keratinocytes.

**Fig. 6 | SDC4 regulates CAR peptide-mediated ARF6 activity. a, b** ARF6 activity (ARF6 GTP) assessed by effector pull-down in HaCaT keratinocytes in the presence or absence of 10 μg/ml CAR or *m*CAR peptide. **a** Representative blots of ARF6 activity during time-course. ARF6 GTP: ARF6 detection in GST-GGA3 pull-down eluate. Total ARF6: ARF6 expression in total cell lysate. Tubulin: total cell lysate loading control. **b** Mean ARF6 activity, relative to total ARF6 ± S.E.M. normalised to 0 min Nil treatment. $N = 3-7$ independent biological replicate experiments (Nil: 0 min $N = 7$, 60 min $N = 3$; CAR: 10–60 min $N = 3$, 120 min $N = 7$; *m*CAR: 30–60 min $N = 3$, 120 min $N = 6$). Datapoints represent ARF6 activity per experiment. Two-way ANOVA with Tukey's multiple comparisons test: Nil 0′ vs CAR 120′ $P = 6.457 \times 10^{-5}$; CAR 120′ vs *m*CAR 120′ $P = 6.732 \times 10^{-6}$; Nil 60′ vs CAR 60′ $P = 4.865 \times 10^{-5}$; CAR 60′ vs *m*CAR 60′ $P = 0.0063$. Holm-Sidak $t$ test: Nil 0′ vs CAR 10′ $P = 0.0004$. **c–e** ARF6 activity in HaCaT cells transfected with control siRNA (CTRL KD), human SDC4-targeting siRNA oligo #1 (SDC4 KD #1) or human SDC4-targeting siRNA oligo #2 (SDC4 KD #2), in the presence or absence of 10 μg/ml CAR or *m*CAR peptide. **c** Representative blots of ARF6 activity during time-course. ARF6 GTP: ARF6 detection in GST-GGA3 pull-down eluate; total ARF6: ARF6 expression in total cell lysate. **d** Mean ARF6 activity, relative to total ARF6 ± S.E.M. normalised to Nil treatment. $N = 3-5$ independent replicate experiments (Ctrl KD: Nil 0 min $N = 5$; CAR 10 min $N = 4$, 30–120 min $N = 5$; *m*CAR 120 min $N = 3$. SDC4 KD #1: Nil 0 min $N = 3$; CAR 10–120 min $N = 3$; *m*CAR 120 min $N = 3$. SDC4 KD #5: Nil 0 min $N = 5$; CAR 10–120 min $N = 5$; *m*CAR 120 min $N = 3$). Datapoints represent ARF6 activity per experiment. Two-way ANOVA with Tukey's multiple comparisons test: CTRL KD Nil 0′ vs CAR 120′ $P = 0.0299$; CTRL KD CAR 120′ vs SDC4 KD #1 CAR 120′ $P = 0.0499$; CTRL KD CAR 120′ vs SDC4 KD #2 CAR 120′ $P = 0.0132$. Holm-Sidak $t$ test: CTRL KD Nil 0′ vs CTRL KD CAR 10′ $P = 0.0012$. Source data are provided in Source Data file. **e** Flow cytometric analysis of cell surface SDC4 in HaCaTs following SDC4 knockdown.

## Cytohesin-2 regulates CAR-dependent ARF6 activity and cell migration

Having established a role for ARF6 and SDC4 in promoting CAR-dependent cell migration, we sought to understand how CAR peptide regulates ARF6 activity. We previously demonstrated that phosphorylation of SDC4 tyrosine-180 (SDC4-$^P$Y180) modulates ARF6 activity[13]. So, using this information, we initiated a preliminary proteomic screen to identify putative ARF6 regulators associated with SDC4. Proteins co-immunoprecipitating with human SDC4 (huSDC4) from SDC4WT, SDC4Y180E (phospho-mimetic) and Syn4Y180L (phospho-null) MEFs were subjected to mass spectrometry and SDC4−/− MEFs were used as a negative control (Fig. 8a; Supplementary Data 1).

ARF GTPases are activated by guanine nucleotide exchange factors (GEFs) and inactivated by GTPase activating proteins (GAPs), which together modulate cycling between GTP- and GDP-loading, respectively. To identify putative SDC4-dependent ARF6 regulatory molecules, the proteomic data were mapped onto the GO Term "Regulation of ARF Protein Signal Transduction" [GO: 0032012]. Five ARF-GEFs were found to co-immunoprecipitate with huSDC4, of which three are reported to modulate ARF6 activity[27]; CYTH2 (also known as Cytohesin-2 or ARNO), IQSEC1 (also known as BRAG2) and CYTH3 (also known as Cytohesin-3 or ARNO3) (Fig. 8a, b).

To prioritise candidate GEFs which might be involved in CAR-mediated wound healing, we analysed single-cell transcriptomic expression in skin wounds[23]. Analysis of scRNA-Seq data revealed elevated levels of CYTH2 expression in keratinocyte populations, which also express high levels of SDC4 and ARF6 (Fig. S8a, b). By contrast, IQSEC1 and CYTH3 exhibited substantially lower levels of expression in keratinocytes, relative to other cell populations (Fig. S8a, b). IHC analysis of day 7 mouse wounds further demonstrated expression of CYTH2 protein in the hyperproliferative epidermis, which also expressed SDC4 and ARF6 (Fig. S8c).

Having established that the ARF-GEF CYTH2 is expressed with SDC4 and ARF6 in wounds, we tested whether CAR peptide influences the ability of CYTH2 to associate with ARF6 in keratinocyte cells. Importantly, CAR peptide induced a striking increase in co-localisation between CYTH2 and ARF6, after 60 min of treatment (Fig. 8c, d). Whereas *m*CAR control peptide had no effect on CYTH2 and ARF6 colocalisation (Fig. 8c, d). CAR treatment also increased the ability of CYTH2 to co-immunoprecipitate with ARF6, while this co-association was unaffected by *m*CAR (Fig. 8e). Interestingly, CAR peptide reduced co-localisation between ARF6 and IQSEC1 (Fig. S8d, e), suggesting that CAR peptide may promote dissociation of IQSEC1 form ARF6 in order to promote binding to CYTH2.

We next analysed whether CYTH2 is required for CAR-dependent ARF6 activation and cell migration. As observed previously, in control conditions, CAR peptide induced ARF6 activation, however, following siRNA-mediated knockdown of the ARF-GEF CYTH2, ARF6 activity was dysregulated and CAR was incapable of inducing ARF6 activation

(Fig. 8f, g). Moreover, while CAR peptide accelerated scratch wound closure and epithelial cell migration in non-targeting siRNA-expressing cells, depletion of CYTH2 inhibited CAR-dependent migration (Fig. 8h–k). Thus, CYTH2 is fundamentally required for CAR-mediated ARF6 activation and for the SDC4- and ARF6-dependent increase in keratinocyte migration triggered by CAR stimulation. Together, these data suggest that CAR peptide, which binds to and is internalised by SDC4, promotes co-association between the ARF-GEF CYTH2 and ARF6 to induce ARF6 activation and drive cell migration.

## CAR peptide requires syndecan-4 to accelerate wound re-epithelialisation

Mechanistically, cell-based analyses suggested that SDC4 is required for internalisation of CAR peptide and that CAR stimulates SDC4-dependent ARF6 activity to control engagement of α5β1 integrin and epithelial cell migration. As our previous experiments suggested that CAR peptide accelerates wound healing in vivo by promoting keratinocyte migration (Figs. 2–4; S1), we sought to determine whether SDC4 is required for CAR peptide to stimulate wound re-epithelialisation in vivo. Skin wounds in SDC4 knockout (KO) and wild-type mice were treated with CAR peptide. Systemic i.v. administration of two daily doses of 3.0 mg/kg CAR or BSA/PBS via tail vein injections was initiated 24 h post-wounding. On day 7 post-wounding, skin wound tissues were collected and histologically analysed. As we have observed previously (Figs. 2, 3), CAR peptide enhanced wound re-epithelialisation of wild-type SDC4-expressing mice. By contrast, wound re-epithelialisation was not accelerated in SDC4 KO mice (Fig. 9a, b, f). Thus, the size of the gap remaining without epidermis (WG$_X$) following CAR treatment was significantly smaller in SDC4 wild-type mice but was not influenced in SDC4 KO mice (Fig. 9a). CAR peptide also increased the incidence of wounds with complete re-epithelialisation from 14% to 41% in SDC4 wild-type mice (Fig. 9b). Moreover, CAR increased the area of HPE tongues by 60% in normal mice but had no effect in SDC4 KO mice (Fig. 9c). As observed previously at 7 days post-wounding (Fig. 3e, f), CAR treatment did not modulate overall wound width (W$_X$), ruling out contraction-driven wound closure, nor the area of granulation tissue relative to control treatment in either wild-type or SDC4 KO mice (Fig. 9d, e).

Finally, we examined homing of CAR peptide to SDC4 WT and KO wounds using FAM-labelled CAR peptide for the final treatment dose. CAR peptide administered 4 h before sacrifice was detected at high levels in SDC4 WT wounds, but not in SDC4 KO wounds (Fig. 9g). In SDC4-expressing mice, CAR peptide was recruited predominantly to the hyperproliferative epidermis, but could also be detected in vascular structures within granulation tissues and in the dermis; areas where SDC4 was also detected in wounds (Fig. 9g). Together these data demonstrate that CAR peptide utilises SDC4 to home to and penetrate wound tissue, but also uses SDC4-dependent signalling to promote wound healing and re-epithelialisation. Thus, CAR appears to be an

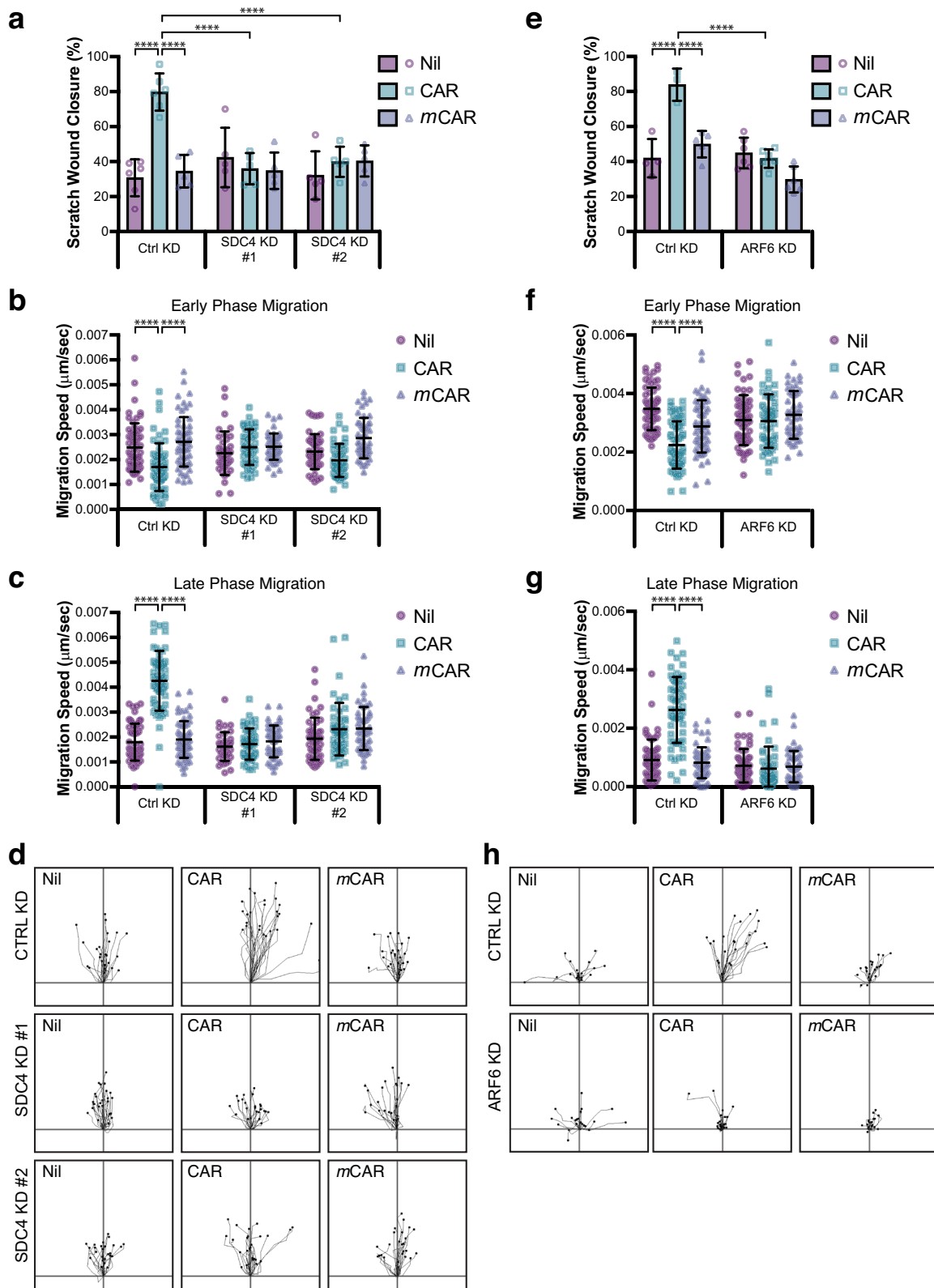

injury-targeting peptide, with potential utility for early intervention in wound healing.

## Discussion

We show here that a homing peptide, named CAR, originally discovered as a wound-homing peptide and used as a delivery vehicle for therapeutics[2–4], also possesses an inherent ability to accelerate wound healing in mice. We further show that CAR peptide acts by activating SDC4-dependent mechanisms to drive epithelial cell migration (Fig. 10).

The CAR peptide is an example of the propensity of in vivo phage display screening for homing peptides to yield peptides that are bioactive, in addition to being able to home to a particular target in the body[6]. Earlier examples of such peptides include the tumour-

**Fig. 7 | CAR promotes SDC4- and ARF6-dependent keratinocyte migration.**
Migration of HaCaT keratinocytes on fibronectin in scratch wound assays, in the presence or absence of 10 µg/ml CAR or *m*CAR peptide. Cells were analysed over 17 h (**a**–**d**) or 20 h (**e**–**h**) by time-lapse microscopy. **a**–**d** HaCaT cells transfected with control siRNA (CTRL KD), human SDC4-targeting siRNA oligo #1 (SDC4 KD #1) or human SDC4-targeting siRNA oligo #2 (SDC4 KD #2). **a** Scratch wound closure, (**b**) speed at early phase of migration (Timepoint 0–5 h), (**c**) speed at late phase of migration (Late: Timepoint 12–17 h), and (**d**) representative migration tracks during late phase migration. See also Supplementary Movie S3 and S4 and Supplementary Fig. S7a, b. Data are representative from one of four independent experiments. Values are means ± S.E.M. All statistical analyses are two-way ANOVA with Tukey's multiple comparisons test. **a** $n = 5$–6 fields of view per condition (Ctrl KD: Nil & CAR $n = 6$; *m*CAR $n = 5$. SDC4 KD #1 & #2: Nil, CAR & *m*CAR $n = 5$); CTRL KD Nil vs CAR $P = 1.383 \times 10^{-7}$; CTRL KD CAR vs *m*CAR $P = 2.239 \times 10^{-6}$; CTRL KD CAR vs SDC4 KD #1 CAR $P = 4.341 \times 10^{-6}$; CTRL KD CAR vs SDC4 KD #2 CAR $P = 2.630 \times 10^{-5}$. **b**, **c** $n = 40$–60 cells per condition. **b** Early phase: CTRL KD Nil vs CAR $P = 9.808 \times 10^{-6}$; CTRL KD CAR vs *m*CAR $P = 1.639 \times 10^{-9}$. (**c**) Late phase: CTRL KD Nil vs CAR $P = 2.520 \times 10^{-11}$; CTRL KD CAR vs *m*CAR $P = 2.520 \times 10^{-11}$. **e**–**h** HaCaT cells

transfected with control siRNA (CTRL KD) or human ARF6-targeting siRNA. **e** Scratch wound closure, (**f**) speed at early phase of migration (Timepoint 0–5 h), (**g**) speed at late phase of migration (Late: Timepoint 15–20 h), and (**e**) representative migration tracks during late phase migration. See also Supplementary Movie S5 and S6 and Supplementary Fig. S7c, d. Data are representative from one of four independent experiments. Values are means ± S.D. All statistical analyses are two-way ANOVA with Tukey's multiple comparisons test. **e** $n = 4$–6 fields of view per condition (Ctrl KD: Nil $n = 4$, CAR $n = 3$; *m*CAR $n = 5$. ARF6 KD: Nil, CAR & *m*CAR $n = 6$); CTRL KD Nil vs CAR $P = 6.312 \times 10^{-6}$; CTRL KD CAR vs *m*CAR $P = 7.806 \times 10^{-5}$; CTRL KD CAR vs ARF6 KD CAR $P = 1.636 \times 10^{-6}$. **f, g** $n = 49$–60 cells per condition (Ctrl KD: Nil & CAR $n = 60$, *m*CAR $n = 59$. ARF6 KD: Nil $n = 58$, CAR & *m*CAR $n = 49$). **f** Early phase: CTRL KD Nil vs CAR $P = 6.82 \times 10^{-13}$; CTRL KD CAR vs *m*CAR $P = 0.0006$; CTRL KD CAR vs ARF6 KD CAR $P = 8.897 \times 10^{-6}$ (**g**) Late phase: CTRL KD Nil vs CAR $P = 4.68 \times 10^{-13}$; CTRL KD CAR vs *m*CAR $P = 4.68 \times 10^{-13}$; CTRL KD CAR vs ARF6 KD CAR $P = 4.68 \times 10^{-13}$ (**a**, **e**) Each data point represents a single field of view; (**b**, **c**, **e**, **f**) Each data point represents an individual cell. Source data are provided as a Source Data file.

penetrating peptides iRGD and LyP-1; iRGD has an inherent anti-metastatic activity[28] and LyP-1 elicits apoptosis in tumour cells and tumour macrophages[29,30]. The reason for this inherent bioactivity is that peptides interact with binding pockets in proteins, and those pockets are generally important active sites (for discussion see[31]). CAR peptide is different from the existing bioactive peptides that bind to proteins in that its target molecule is a carbohydrate, the heparan sulfate glycosaminoglycan (GAG) chains of HSPGs[3,4]. CAR contains a classical heparin-binding domain (HBD) that has high homology with the HBD of bone morphogenetic protein-4, and we have previously shown that CAR binds to heparan sulfate and heparin[3,4]. HBD-GAG interactions are thought to be mostly charge mediated, i.e., dependent on attraction between basic residues in the HBD and sulfated sugar residues in the GAG[8,9,32,33]. HSPGs are ubiquitous, however the selective homing of CAR to angiogenic vessels and wound tissue suggests additional elements to the CAR specificity, such as a GAG sulfation pattern that creates a molecular signature characteristic of regenerating tissues. This scenario is supported by the recent demonstration that SDC4 expression is very low or absent from normal quiescent blood vessels, but it is the key syndecan family member that exhibits enhanced expression during pathological angiogenesis[34]. Furthermore, it was also shown that owing to differences in the heparan sulfate chains of SDC2 and SDC4 (defined by a unique protein sequence in SDC2 ectodomain), the heparin-binding growth factor, vascular endothelial growth factor-A (VEGFA), binds only to SDC2 and not to SDC4[35]. Thus, two factors, distinct GAG chains on specific HSPGs, and selective overexpression of SDC4 in migrating epithelia and angiogenic vessels, may contribute to the selectivity of CAR to wounds.

Thetargeted, organ- and cell-specific mode of action of CAR makes it possible to use systemic administration of the peptide to accelerate wound healing, which circumvents limitations of local treatments, such as difficulty in maintaining the activity of local agents in the wound environment. Systemic treatment is also advantageous when the injured site cannot be accessed by topical application, when multiple tissues are injured simultaneously, or if a certain phase of healing is desired[4]. Our results only address the treatment of skin wounds, so it will now be important to determine whether the beneficial activities of CAR extend to other tissues. It is known that CAR homes to injuries in tendon[3] as well as to diseased tissue in pulmonary arterial hypertension, bronchopulmonary dysplasia, cancer (human tumour xenografts), aortic aneurysms, retinopathies, muscular dystrophies and myocardial infarction[16–18,36–41]. In line with CAR peptide homing to tissue injuries and these diseases, the upregulation of SDC4 expression has been described in all these instances[9,22,34,39,42,43].

Several lines of evidence from our study indicate that CAR promotes wound healing through selective engagement of SDC4. First,

wounds in SDC4 knockout *mice* were impervious to the CAR-mediated effect. Second, our mechanistic data shows that SDC4 is required for internalisation of CAR peptide and that CAR modulates SDC4-dependent ARF6 activity to drive epithelial cell migration. Third, CAR peptide induced SDC4- and ARF6-dependent redistribution of α5β1 integrin, indicating that CAR regulates cell migration by orchestrating integrin engagement. Finally, SDC4 is expressed de novo on keratinocytes in the migrating epidermis (epidermal tongues), and re-epithelialisation is the main aspect of wound healing that is affected by CAR.

We also found that although the size of the granulation tissue was similar in the CAR-treated and control wounds, there was a striking difference in the number of myofibroblasts in the two treatment groups; the CAR-treated wounds almost completely lacked myofibroblasts, whereas myofibroblasts were abundant in the control-treated wounds. Myofibroblast-driven contraction of loose granulation tissue into a scar completes wound healing, but this process comes at a price, with formation of a permanent scar[44]. As there are fewer myofibroblasts in CAR-treated wounds, CAR may also reduce scarring. This scenario is supported by recent findings that SDC4 expression suppresses the development of fibrosis in fibrotic disease models[45–47]. However, this remains to be studied because the 10-day observation period used to assess re-epithelialisation in this study was too short to assess permanent scar formation. In this regard, it will be important to determine whether the reduced number of myofibroblasts at day 10, is due to the earlier resolution of the wounds following CAR treatment, or because of inherent anti-fibrotic activity.

The SDC4-dependent cell migration pathway that underpins the mechanistic basis of CAR-accelerated wound healing, has been characterised in detail[11–13,20,48]. While many ECM proteins contain HBDs, fibronectin is the main endogenous SDC4 ECM ligand that activates these pro-migratory pathways[7,11]. Plasma fibronectin is a major component of the blood clot that forms immediately after wounding and fibronectin is also abundantly expressed by fibroblasts in granulation tissue[49]. Although plasma fibronectin is not essential for wound healing[50], cellular fibronectin is indispensable and lack of fibronectin extra domain A (EDA) leads to selective defects in wound re-epithelialization[51]. Thus, it is thought that fibronectin provides a bed/scaffold for migrating epidermis to close the wound[51]. Yet our analysis of wound tissue indicates that very few migrating keratinocytes are in contact with fibronectin during wound repair. Thus, it is probable that a large proportion of keratinocytes express unligated SDC4 that can receive an additional migration-promoting stimulus from CAR. Thus, we propose a model whereby, in the early stages following wounding, elevated SDC4 expression promotes CAR targeting to the site of injury, but also provides a reservoir of unengaged receptors that are available

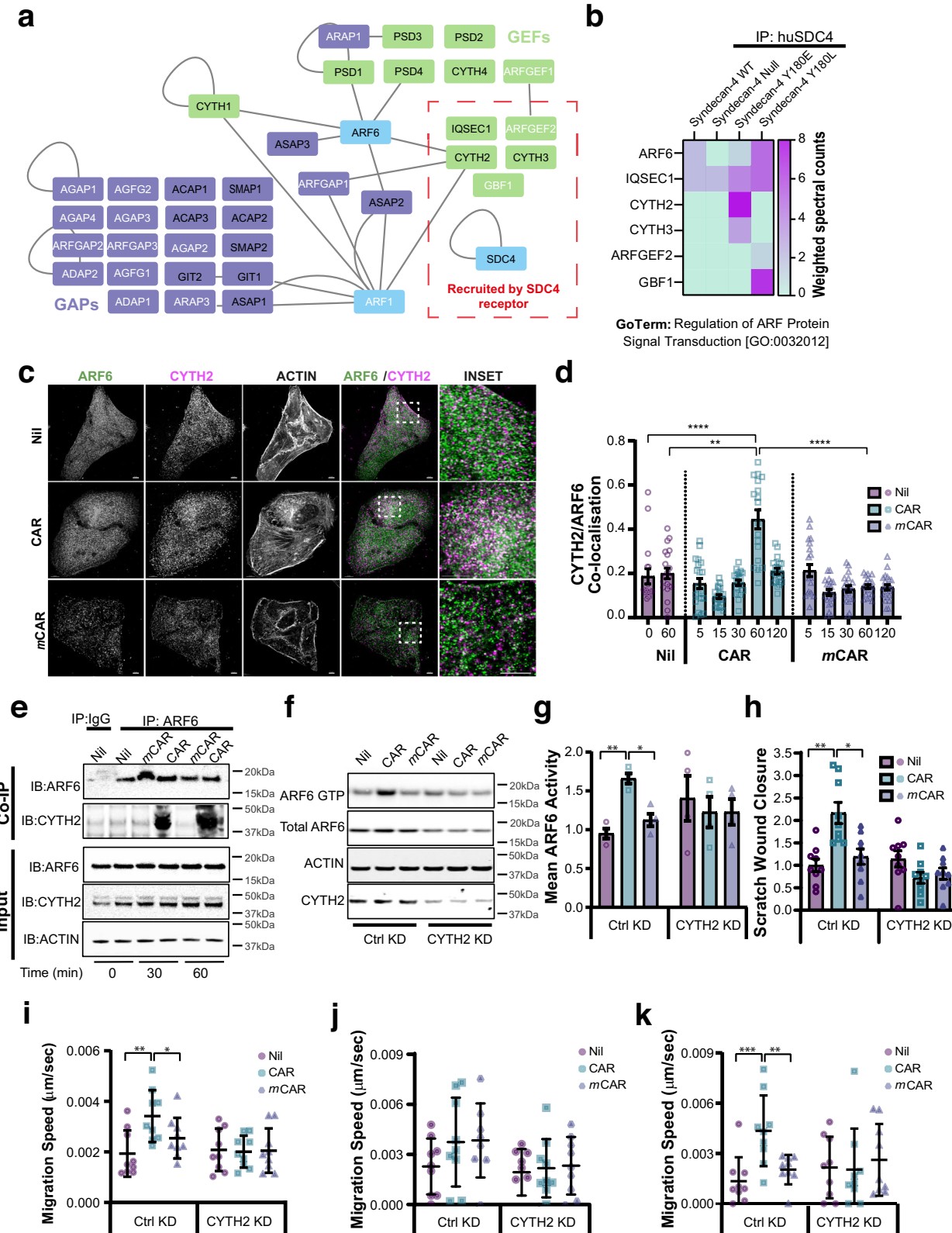

to be bound by the peptide to accelerate the endogenous wound healing response selectively on the epidermis.

Mechanistically, our data indicate that ARF6 is the downstream effector of SDC4 signalling triggered by CAR treatment, via the modulation of CYTH2 recruitment, to drive epithelial cell migration. Accumulating evidence suggests that integrin recycling plays a key role in cell migration[52,53]. As SDC4 regulates ARF6 to co-ordinate

heterodimer-specific recycling of integrins to the plasma membrane[13], it is likely that CAR peptide regulates cell migration and wound healing by driving recycling of integrins. Indeed, the SDC4- and CYTH2-dependent activation of ARF6 induced by CAR, and the SDC4- and ARF6-mediated redistribution of α5β1 in response to CAR stimulation, are consistent with CAR regulating cell migration and wound healing by orchestrating α5β1 engagement. However, as SDC4 is also a growth

**Fig. 8 | CYTH2 regulates CAR peptide-mediated ARF6 activity. a, b** Proteomic analysis of ARF regulatory molecules co-immunoprecipitating with human SDC4 (huSDC4) from Syn4WT, Syn4-/-, Syn4Y180E and Syn4Y180L MEFs. **a** Protein-protein interaction network of molecules in the GO Term "Regulation of ARF Protein Signal Transduction" [GO:0032102]. GEFs: green nodes; GAPs: purple nodes; Blue nodes: proteins not in GO:0032012 (ARF6, ARF1 GTPases and SDC4 bait protein); Edges (grey lines): known protein-protein interactions; Black labels: proteins with reported ARF6 activity modulation properties[27]; white labels: proteins not reported to modulate ARF6 activity[27]. Red dashed box: proteins co-immunoprecipitating with huSDC4. **b** Heatmap displaying proteins within GO:0032012 co-immunoprecipitating with huSDC4. Colour-coding indicates enrichment levels (weighted spectral counts). **c, d** Quantitative analysis of CYTH2/ARF6 co-localisation following CAR peptide stimulation. **c** Subcellular distribution of CYTH2 (magenta) and ARF6 (green) following 60-min treatment 10 μg/ml CAR, *m*CAR or vehicle control. Dashed boxes: inset regions. Scale bars: 5 μm (main images); 2 μm (insets). **d** Pearson's coefficient of CYTH2 and ARF6 co-localisation ± S.E.M. following 0–120-min treatment with 10 μg/ml CAR or vehicle control. Datapoints represent mean Pearson's coefficient of CYTH2 and ARF6 co-localisation per image. *N* = 3 independent replicate experiments with 17–22 images analysed per condition. Kruskal-Wallis test with Dunn's multiple comparisons test: Nil 0' vs CAR 60' $P = 7.385 \times 10^{-5}$; Nil 60' vs CAR 60' $P = 0.0019$; CAR 60' vs *m*CAR 60' $P = 3.828 \times 10^{-6}$. **e** CAR peptide promotes association of CYTH2 with ARF6. Immunoprecipitation of ARF6 following 0, 30 or 60 min CAR or *m*CAR treatment.

immunoprecipitation with rabbit anti-ARF6 (IP: ARF6); or non-immune rabbit IgG (IP: IgG). Immune complex-associated CYTH2 and ARF6 detected by western blot. **f, g** ARF6 activity in CTRL KD or CYTH2 KD keratinocytes following 120 min in presence or absence of CAR or *m*CAR. *N* = 4 Independent biological replicate experiments. **f** Representative blots of ARF6 GTP (GST-GGA3 pull-down eluate); Total ARF6: ARF6 expression in total cell lysate. Actin detection in TCL acts as a loading control. CYTH2 detection in TCL demonstrates level of siRNA-mediated knockdown. **g** Mean ARF6 activity relative to total ARF6 ± S.E.M. normalised to Nil treatment in Ctrl KD cells. N = 4 independent replicate experiments. Datapoints represent ARF6 activity in each experiment. Brown–Forsythe and Welch ANOVA test for multiple comparisons assuming non-equal variance: CTRL KD Nil vs CAR $P = 0.0027$; CTRL KD CAR vs *m*CAR $P = 0.0213$. **h–k** Migration of control (CTRL KD) or human CYTH2 knockdown (CYTH2 KD) HaCaTs in presence or absence 10 μg/ml CAR or *m*CAR peptide. **h** Scratch wound closure relative to untreated Ctrl KD cells, **(i)** mean migration speed throughout timelapse, **j** speed at early migration phase (0–5 h), **k** speed at late migration phase (12–17 h). Migration data are means ± S.E.M from three independent experiments in triplicate. Statistical analyses are two-way ANOVA with Tukey's multiple comparisons test: **h** Scratch wound closure: CTRL KD Nil vs CAR $P = 3.331 \times 10^{-5}$; CTRL KD CAR vs *m*CAR $P = 0.0005$; **i** Total migration speed: CTRL KD Nil vs CAR $P = 0.017$; CTRL KD CAR vs *m*CAR ns; **k** Late phase: CTRL KD Nil vs CAR $P = 0.0039$; CTRL KD CAR vs *m*CAR $P = 0.0317$. Source data are provided as a Source Data file.

---

factor co-receptor[7], CAR-dependent ARF6 activation may also regulate growth factor receptor trafficking to co-ordinate cell migration and wound healing[24]. A significant level of crosstalk exists between adhesion receptors and growth factor receptors and this interplay has critical roles in cell migration[54–56]. Reciprocal, and mutually regulatory, trafficking mechanisms are one of the major mechanisms by which adhesion receptor and growth factor receptor signals are integrated[52,54,55]. Thus, as syndecans and ARF6 regulate trafficking of both integrins and growth factor receptors[11,13,24,57], it is possible that CAR may co-ordinate ligand engagement and signalling of multiple pro-migratory receptors.

Recent studies report the use of fibrin-conjugated peptides as a locally administered biomaterial to enhance wound healing[58]. The authors attributed the effects of these biomaterials on wound healing to growth factor retention afforded by the growth factor binding to the HBD of these peptides in the wounds[58]. Since these peptides contain HBD and bind to syndecans[58,59], there is some similarity to our work. However, we use soluble peptide, that accumulates in the wounds by homing and cell penetration and is not covalently anchored to ECM, therefore it is unlikely that growth factor retention is a significant part of its mode of action. Moreover, our treatment is systemic which offers considerable advantages over local administration. Furthermore, these topically administered, syndecan-binding peptides are also functionally different from CAR in that their activities include, for example, stimulation of proteolytic activity by matrix metalloproteinases[60].

Finally, the classical view of the role of heparan sulfate in the biology of heparin-binding growth factors is that their binding to HSPGs merely provides a way to concentrate and present growth factors to a signalling receptor, such as a receptor tyrosine kinase[32,33]. As CAR peptide stimulates wound healing through SDC4-/HSPG-dependent signalling, it may be that heparin-binding growth factor signalling is more complex than currently thought, and that HSPGs, in addition to concentrating and presenting growth factors, also act as signalling receptors in concert with conventional growth factor receptors. Moving forward, it will be important to determine how integration of cell-matrix adhesion receptor and growth factor receptor functions are integrated to control migratory behaviour and wound repair, at both the sub-cellular and tissue level.

Defective wound healing has been reported in SDC4 knockout mouse as well as in mice with a conditional, keratinocyte-specific ARF6

deletion[8,9,61,62]. These studies, when considered alongside our data, highlight the key roles that both SDC4 and ARF6 play in wound healing and suggest that pharmaceutical manipulation of these pathways to promote wound healing may be therapeutically tractable. Thus, CAR peptide may provide a way of enhancing wound healing by systemic treatment, and perhaps tissue regeneration in general. Indeed, it has been shown that SDC4 also plays key roles in skeletal muscle regeneration[63,64], epithelial regeneration in experimental colitis models[65] and fracture and cardiac repair[39,66]. We have not seen any obvious toxicities in any of the many mice we have treated with CAR over extended periods of time. Moreover, CAR is active in human cells and tissues: the human keratinocyte cell line we used in this study responded to CAR with increased migration, and previous studies have shown that CAR binds to and penetrates human endothelial cells and homes to human tumour xenografts in vivo[16]. These features bode well for the translational prospects of CAR.

## Methods
### Mouse husbandry and derivation
Mice were housed in the same animal rooms in pathogen-free animal facilities. Animals were kept in a 12-h light/dark cycle, with controlled humidity and temperature. They were fed with standard laboratory pellets and water *ad libitum*. All animal experiments were performed in accordance with protocols approved by the National Animal Ethics Committee of Finland, the institutional animal care and use committees of the Sanford Burnham Prebys Medical Discovery Institute (La Jolla, CA, USA) and the University of California at Santa Barbara (Santa Barbara, CA, USA).

In this study BALB/c (Janvier Labs; Harlan Laboratories) and SDC4 KO mice were used. The generation of SDC4 KO mouse has been described in detail elsewhere[67]. SDC4 KO mice were obtained from Dr. Mark Bass (University of Sheffield, Sheffield, UK). Before initiating experiments, SDC4 KO *mice* were re-derived, backcrossed eight generations with C57BL/6 strain (Janvier Labs) to obtain SDC4 expressing (wild-type, WT) and SDC4 KO strains in the same genetic background (littermates). Then homozygous SDC4 KO animals were bred. The genotype was determined in each animal by PCR.

### Peptide synthesis
CAR and modified CAR peptides (CAR: CAR*SKNKDC*; *m*CAR: CAQSNNKDC) were synthesised with an automated peptide

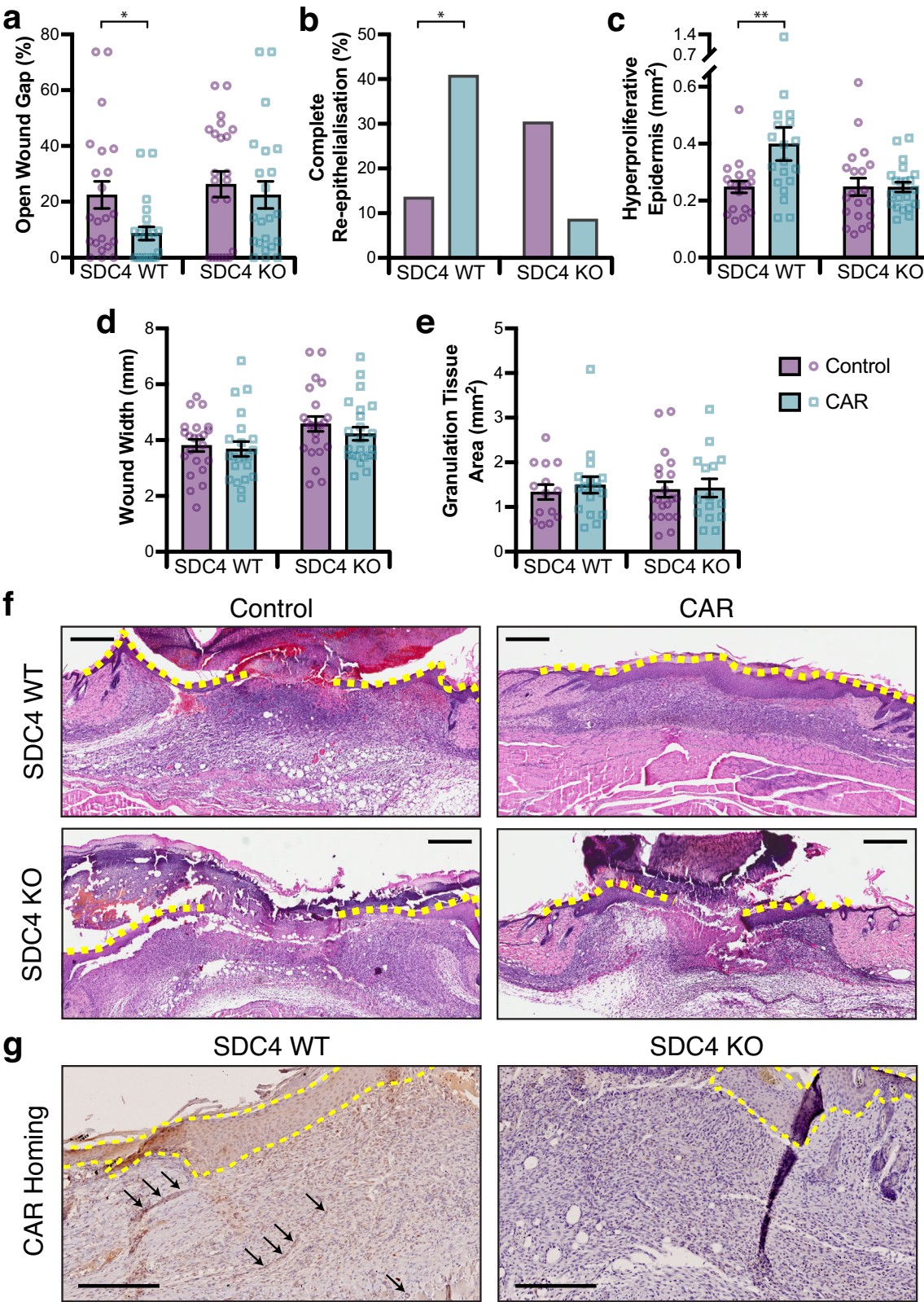

synthesiser by using standard solid phase fluorenylmethoxycarbonyl chemistry. During synthesis, some batches of peptides were labelled with fluorescein amitide (FAM) using an amino-hexanoic acid spacer for use in imaging studies[18]. The peptides were stored in −20 °C, dissolved in PBS immediately before experiments, kept on ice, shielded from light and used fresh (within the same day).

**Wound healing model and treatment schedule**

8–10-week male mice (BALB/c, or SDC4 KO C57BL/6 and SDC4 WT C57BL/6 littermates), weighing 23–28 g, were used in wound studies. Mice were anaesthetised with 4% isoflurane and 1.5 l/min of oxygen, and the anesthesia was maintained at ≈1.5% isoflurane at 1 l/min of oxygen. Skin was shaved, cleaned, and disinfected with betadine and

**Fig. 9 | SDC4 is required for CAR peptide induced wound re-epithelialisation and CAR homing.** Male SDC4 WT and KO mice with full thickness skin excision wounds were treated either with systemically i.v. administered CAR peptide or BSA/PBS (control) as described in methods, wounds were harvested on day 7. **a** Gap between the epidermal tongues (WG$_X$: area of open wound without re-epithelialisation) represented as percentage (%) of original wound size. SDC4 WT: CAR vs. Control $P = 0.05$. **b** Percentage of completely re-epithelialised wounds. SDC4 WT: CAR vs. Control $P = 0.042$. **c** Area of hyperproliferative epidermal tongues. SDC4 WT: CAR vs. Control $P = 0.0069$. **d** Overall wound width, indicative of contraction. **e** Cross-sectional area of granulation tissue quantified by examining two histological HE-stained sections from each wound. **f** Representative histological HE-stained pictures of the wounds treated with CAR peptide and BSA/PBS (Control) collected on day 7 are shown for wound re-epithelialisation. The epidermal tongues are marked with yellow dashed lines. Six animals, each with four wounds, in every treatment group. Scale bars: 800 µm. Values are mean ± S.E.M. Each data point represents an individual wound. $N = 24$ wounds in each treatment group (**a–e**). Wilcoxon Mann–Whitney test with tie correction and Bonferroni post-hoc test (two-sided) (**a, c, d** and **e**) and Pearson´s Chi-square test (without continuity correction) with post-hoc test Fisher exact test (two-sided) (**b**). Source data are provided as a Source Data file. **g** CAR peptide homing. Representative micrographs of CAR-peptide distribution in skin wound sections are presented for SDC4 WT and SDC4 KO groups, following administration of fluorescein-labelled CAR-peptide for 4 h prior to wound harvesting. Peptide localisation determined by anti-fluorescein immunodetection. Representative micrographs of CAR-peptide distribution in skin wound sections are presented for SDC4 WT and SDC4 KO groups. Arrows indicate CAR homing to blood vessels. $n = 12$ individual wounds; three animals with four wounds. Scale bar: 250 µm.

70% alcohol. Treatment trials were conducted on mice that had four circular, 6 mm diameter, full-thickness excision wounds (including panniculus carnosus muscle) in the dorsal skin (two on both sides of the dorsal mid-line)[4]. The wounds were first marked by a 6 mm biopsy punch and then cut with scissors. All skin wounds were left uncovered without a dressing. The animal welfare was monitored every 12 h throughout the experiments.

The three systemic, randomised peptide treatment trials were carried out in in BALB/c mice, where approximately 60% and 40% of wound closure is driven by re-epithelialisation and panniculus carnosus-driven wound contraction, respectively[19]. The systemic peptide treatment trial on SDC4 WT and KO mice was carried on C57BL/6 littermates. The wounded mice were randomized to different treatment groups (by housing cages). The systemic peptide treatments were started 24 h after wounding and consisted of two daily tail vein injections under anesthesia. When extravasation was identified during the tail vein-injection, the injection was stopped and the rest of the injection was administrered s.c. on the ventral side of the body to guarantee systemic administration. This systemic peptide injection schedule continued until the sacrifice of the animals. The dose for CAR and *m*CAR peptides was 3.0 mg/kg (approximately 62.5 µg per animal) per injection in 100 µl PBS on the basis of previous CAR peptide treatment studies[16–18]. BSA (62.5 µg) was used as a control protein in PBS injections. At the end of the treatment period, the animals were sacrificed with carbon dioxide, the wounded tissue collected and processed for further analyses.

## Morphological assessment of wound closure

After the surgery and daily after that until the sacrifice, the wounds were photographed digitally. Two 2 × 2 cm cardboard squares were placed on both sides of the animal under anesthesia to adjust the digital pictures taken from various distances in relation to wounds and the total area of a wound was measured and analysed from digital photographs (coded) using ImageJ software (NIH) by manually drawing the edges of each individual wound as described previously[4]. The investigator was blinded to the treatment group allocation.

## Histology

Mice were euthanised and perfused by intra-cardial injection of 4% paraformaldehyde (PFA). Excision of a rectangular section of skin containing all wounds, as well as underlying skeletal muscle, was performed to ensure the uninterrupted wound[3,4,16]. The "whole-mounted" sections were immobilised on filter paper, immersed in 4% PFA for additional overnight fixation, and washed with physiological saline. Thereafter, the wounds were bisected, dehydrated, and embedded in paraffin. Longitudinal sections (6 µm) from the middle of the wound were stained with hematoxylin/eosin (HE) and/or processed for immunohistochemistry.

## Immunohistochemistry

Immunohistochemical (IHC) staining was performed on 6 µm thick paraffin sections. IHC was carried out using appropriate antigen retrieval method for each staining, essentially as described previously[4]. For double-IHC, antigen retrieval methods for each antibody were tested to identify the ideal method. The following primary antibodies were used for IHC (according to the manufacturer's instructions): 1:200 rat anti-Ki67 (TEC-3; #M7249, Bethyl Laboratories), 1:50 rat anti-CD31 (MEC 13.3; #550274, BD Biosciences), 1:50 rat anti-F4/80 (BM8; #MF48000, Invitrogen), 1:100 rabbit anti-cytohesin-2 (N7; generous gift from Dr. H. Sakagami)[68], 1:400 rabbit anti-Arf6 (#PA1-093, Invitrogen), 1:100 rabbit anti-α-smooth muscle actin (α-SMA) (#ab5694, Abcam), 1:100 rat anti-syndecan-4 (KY/8.2; #550350, BD Biosciences), 1:50 rabbit anti-cytokeratin 17 (#ab53707, Abcam), 1:100 rabbit anti-fibronectin (#ab2413, Abcam) and 1:200 rabbit anti-fluorescein (#71-1900, Invitrogen). The blocking reagents used for IHC were S2O23 REAL and S0809 Antibody Diluent (Dako). The horseradish peroxidase (HRP) conjugated secondary antibodies used were anti-rat Histofine (undiluted; #414311F, Nichirei Bio) for CD31, syndecan-4 and F4/80, goat anti-rabbit (1:200; #P0448, Dako) for Ki-67, cytohesin-2, fibronectin, fluorescein, Arf6 and Rabbit-on-Rodent HRP Polymer anti-rabbit (undiluted; #RMR622H, BioCare Medical) for α-SMA, followed by peroxidase reactive chromogen K3465 DAB (Dako). The samples were mounted with Vectashield mounting medium (Vector Laboratories). Double-immunohistochemistry was carried out sequentially using Vectastain Elite ABC-HRP Kit (Vector Laboratories) in combination with immuno-peroxidase polymer with Vina Green Chromogen Kit (BioCare Medical) as described previously[69]. Harris's haematoxylin was used as a counterstain.

IHC-based peptide homing studies in SDC4 WT and SDC4 KO mice were carried out as described, however, the final treatment 4 h before sacrifice on day 7 post wounding was performed with 3.0 mg/kg (approximately 62.5 µg per animal) CAR-FAM peptide, instead of CAR. Peptide distribution in wound tissue was detected using anti-fluorescein primary antibody.

## Quantitative analysis of histology and immunohistochemistry

Two HE-stained sections from the middle of each wound were quantitatively evaluated, and the average of the two values was used as one value for each wound in the analysis. From all wound data belonging to the same treatment and time-point group the average mean was plotted. When the wound heals, new epidermis grows from both sides of the wound edges and is termed hyperproliferative epithelial tongues. The area of epithelial tongues as well the gap between the two epithelial tongues of a wound (a measure of re-epithelialisation) was measured as described by ref. 19. Schematic representation in Fig. 2a. WG$_X$: Gap between the two epithelial tongues on day X post-wounding. E1$_X$ + E2$_X$: Hyperproliferative epidermis (HPE) length was used to calculate HPE area per wound. W$_X$: As a readout of wound contraction, the

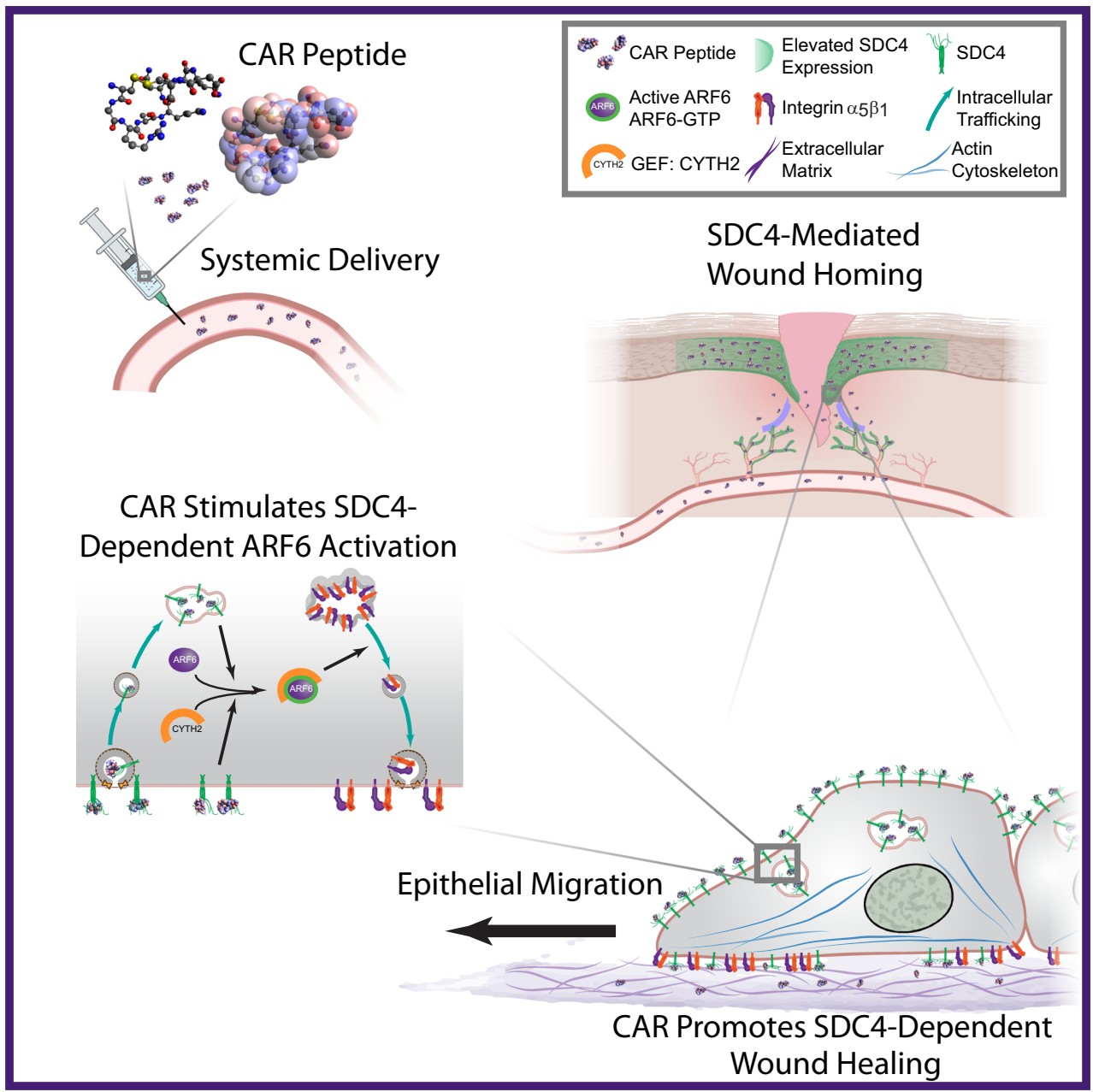

**Fig. 10 | Systemically administered wound-homing peptide accelerates wound healing by modulating SDC4 function.** Schematic diagram highlighting proposed mechanism of action of CAR peptide. Systemically administered CAR peptide associates with the HSPG SDC4, which is restricted to epidermis and blood vessels in mouse skin wounds. CAR induces SDC4-dependent activation of the small GTPase ARF6, via the guanine nucleotide exchange factor CYTH2, to promote SDC4-, ARF6- and CYTH2-mediated keratinocyte migration and endogenous re-epithelialisation and wound repair mechanisms.

total size/width of the entire wound was determined on day X by the distance between the closest hair follicles on each side of the wound. Depending on the experimental setup for histological analysis, day X corresponded to day 5, 7 or 10 post-wounding. The area of granulation tissue was determined by drawing its edges using free-hand tool. Image analysis and quantification of histological and IHC parameters were performed using Spectrum digital pathology system using the Aperio ScanScope CS and XT systems (Aperio, Leica Biosystems) as described previously elsewhere[4,69]. Slides were viewed and analysed with the Aperio ImageScope software (Leica Biosystems). The regions of interests were recorded for each wound. All quantified histochemical analyses (Ki-67, CD31, F4/80, α-SMA) were performed according to the protocols used to establish these algorithms for each respective staining[4,69]. Analysis of Ki67-positive cells in the epidermis extended between 1st hair follicles, to ensure inclusion of the whole activated epidermis, including the proliferative zone. The same microscope settings and downstream image processing parameters were used for all samples within each individual staining group. All animal experiments, including morphological and histological analyses were carried out by adhering to ARRIVE 2.0 guidelines[70].

## Cell culture

The HaCaT spontaneously immortal keratinocyte cell line was obtained from Dr Hong Wang (QMUL, UK) and maintained in Dulbecco's Modified Eagles Medium (DMEM, Sigma-Aldrich) supplemented with 10% foetal bovine serum (FBS). Immortalised Im+, Syn4WT, Syn4Y180E, Syn4Y180L and Syn4-/- mouse embryonic fibroblasts (MEFs)[20], were grown at the large T-antigen permissive

temperature of 33 °C in DMEM supplemented with 10% FBS, 2 mM L-glutamine, and 20 μ/ml IFNγ (Sigma-Aldrich). These cells were grown in the presence of IFNγ to maintain immortalisation during culture. Syn4WT, Syn4Y180E and Syn4Y180L MEFs were generated by retroviral transduction of HA-tagged wild-type human SDC4, or HA-tagged Y180E or Y180L mutated human SDC4, constructs into SDC4 KO MEFs, as described previously[13,20]. Large T-antigen immortalised human dermal fibroblasts (TIFs) were obtained from Dr Patrick Caswell (University of Manchester, UK) and grown in DMEM supplemented with 15% FBS and 2mM L-glutamine at 37 °C.

### Immunofluorescence: peptide internalisation
Cells were plated on sterile coverslips in full growth medium for 24 h, prior to treatment with 10 μg/ml FAM-conjugated CAR peptides (CAR-FAM). Following 8 h incubation, cells were fixed with 4% (wt/vol) paraformaldehyde, permeabilised for 2 min at RT with 0.5% (wt/vol) TritonX-100 in PBS⁻ and blocked (0.1% BSA/0.1% sodium azide in PBS⁻ (0.1/0.1 buffer)). Cells were stained for actin with AlexaFluor-594-conjugated phalloidin (1:400; #A12381, Invitrogen) and/or SDC4 with 5 μg/ml rabbit anti-SDC4 primary antibody (#3644, BioVision) and AlexaFluor-647-conjugated AffiniPure Donkey anti-Rabbit secondary antibody (1:400, 3.75μg/ml; #711-605-152, Jackson ImmunoResearch). Cells were mounted in ProLong Gold (Invitrogen) and immunofluorescent images were acquired on an Olympus IX71 using DeltaVisionRT software, Olympus 60x/NA1.40 Plan Apo objective and Coolsnap HQ camera. The same microscope settings were used to acquire all images within *each* experiment. The same ImageJ settings were applied for all conditions within a single experiment.

### Immunofluorescence: CAR-Induced cytoskeletal reorganisation
To test the role of CAR stimulation on focal adhesion and cytoskeletal organisation, in the absence of syndecan-4 engagement, cycloheximide-treated cells were spread on sterile coverslips coated with 10 μg/ml central cell-binding domain of fibronectin (Fn6-10/50 K) for 120 min, as described previously[11,13,20], then stimulated with 10 μg/ml CAR peptide, *m*CAR peptide or vehicle control for 30 or 120 min before fixation. Cells were fixed with 4% (wt/vol) paraformaldehyde, permeabilised for 2 min at RT with 0.5% (wt/vol) TritonX-100 in PBS⁻ and blocked in 0.1/0.1 buffer. MEFs were stained for α5 integrin using 10 μg/ml anti-*mouse* CD49e primary antibody (5H10-27(MFR5); #553319, BD Biosciences) and AlexaFluor-488-conjugated AffiniPure Donkey anti-Rat secondary antibody (1:400, 3.75μg/ml; #712-545-153, Jackson ImmunoResearch) and stained for actin with AlexaFluor-594-conjugated phalloidin (1:400; #A12381, Invitrogen). TIFs were stained for α5 integrin using 10 μg/ml anti-human α5 integrin primary antibody (mab11; purified in-house from hybridoma) and AlexaFluor-488-conjugated AffiniPure Donkey anti-Rat secondary antibody (1:400, 3.75μg/ml; #712-545-153, Jackson ImmunoResearch) and stained for actin with AlexaFluor-594-conjugated phalloidin (1:400; #A12381, Invitrogen). Immunofluorescent images were acquired on an Olympus IX71 using DeltaVisionRT software, 60x/NA1.40 Plan Apo or 40x/NA0.85 Uplan Apo objectives and Coolsnap HQ camera. The same microscope settings were used to acquire all images within each experiment. The same ImageJ settings were applied for all conditions within a single experiment.

HaCaT cells were plated on fibronectin in the presence of 10% FBS for at least 16 h and serum-starved for 4 h prior to treatment with 10 μg/ml CAR peptide or vehicle control for 0, 5, 15, 30 or 60 min before fixation. Cells were fixed with 4% PFA for 10 min, washed once with 0.1% sodium azide and twice with PBS⁻. Following permeabilisation with 0.1% (v/v) Triton X-100 for 10 min and blockade with 2% BSA in PBS⁻ for 1 h, cells were incubated with were stained for α5 integrin using 1 μg/ml rat anti-human α5 integrin monoclonal antibody (mab11; purified in-house from hybridoma) and paxillin using 0.5 μg/ml mouse anti-paxillin monoclonal antibody (349; #610051, BD Biosciences).

Primary antibodies were detected using AlexaFluor-647-conjugated AffiniPure Donkey anti-Rat (1:400, 3.75 μg/ml; #712-605-153, Jackson ImmunoResearch) and AlexaFluor-594-conjugated AffiniPure Donkey anti-Mouse (1:400, 3.75 μg/ml; #715-585-151, Jackson ImmunoResearch) and actin was stained with AlexaFluor-488-conjugated phalloidin (1:400; #A12379, Invitrogen). Samples were mounted using Prolong Gold anti-fade (Invitrogen) and imaged by confocal microscopy using a Zeiss LSM900 with Airyscan2 system with a 63x/1.4 oil objective (voxel size 0.09 × 0.09 × 0.2 μm). The same microscope settings were used to acquire all images within each experiment. The area of α5 integrin adhesions per cell, in a single z-plane at the cell-matrix interface, was calculated using the Particles Analysis plugin in Fiji.

### Immunofluorescence: ARF6/GEF co-localisation analysis
To analyse CAR-dependent changes in co-localisation between ARF6 and CYTH2 or IQSEC1, HaCaT cells were plated on fibronectin in the presence of 10% FBS for at least 16 h and then serum-starved for 4 h prior to treatment with 10 μg/ml CAR peptide, *m*CAR peptide or vehicle control for 0, 5, 15, 30, 60 or 120 min before fixation. Cells were fixed with 4% PFA for 10 min, washed once with 0.1% sodium azide and twice with PBS⁻. Following permeabilisation with 0.1% (v/v) Triton X-100 for 10 min and blockade with 2% BSA in PBS⁻ for 1 h, cells were incubated with either 5 μg/ml rabbit anti-ARF6 pAb (#PA1-093, Invitrogen) and 5 μg/ml mouse anti-CYTH2 mAb (10A12; #MA1-061, Pierce), or 10 μg/ml mouse anti-ARF6 mAb (3A-1; #sc-7971, Santa Cruz) and 5 μg/ml rabbit anti-IQSEC1 pAb (#PA5-38019, Invitrogen), in PBS⁻ containing 0.5% (w/v) BSA and 0.05% TritonX-100 for 1 h. Primary antibodies were detected using 1:400 AlexaFluor-488 and −647-conjugated species-specific secondary antibodies and actin was detected using AlexaFluor-594-conjugated phalloidin (1:400; A12381, Invitrogen). Cells co-stained for CYTH2 and ARF6 were incubated with AlexaFluor-488-conjugated AffiniPure Donkey anti-Rabbit (1:400, 3.75 μg/ml; #711-545-152, Jackson ImmunoResearch) and AlexaFluor-647-conjugated AffiniPure Donkey anti-Mouse (1:400, 3.75 μg/ml; #715-605-151, Jackson ImmunoResearch) secondary antibodies. Whereas Cells co-stained for IQSEC1 and ARF6 were incubated with AlexaFluor-488-conjugated AffiniPure Donkey anti-Mouse (1:400, 3.75 μg/ml; #715-545-151, Jackson ImmunoResearch) and AlexaFluor-647-conjugated AffiniPure Donkey anti-Rabbit (1:400, 3.75 μg/ml; #711-605-152, Jackson ImmunoResearch) secondary antibodies. All steps were performed at room temperature (RT). Samples were mounted using Prolong Gold anti-fade (Invitrogen) and imaged by confocal microscopy using a Zeiss LSM900 with Airyscan2 system with a 63x/1.4 oil objective (voxel size 0.09 × 0.09 × 0.2 μm). For co-localisation analysis, images were first deconvolved using Huygens professional software (Scientific Volume Imaging). Pearson's coefficient of co-localisation for the region of interest (ROI) was calculated using the co-localisation module of IMARIS 9 software (Oxford Instruments).

### Flow cytometry
To detect levels of cell surface SDC4 expression, cells were detached using enzyme-free Hanks'-based cell dissociation buffer (Invitrogen) and washed with 0.1/0.1 buffer. Cells were incubated with 10 μg/mL mouse anti-Human SDC4 primary antibody (5G9; #sc-12766, Santa Cruz) for 30 min at 4 °C in 0.1/0.1 buffer at 4 °C, washed three times, and incubated with AlexaFluor-488-conjugated AffiniPure Donkey anti-Mouse secondary antibody (1:400, 3.75 μg/ml; #715-545-151, Jackson ImmunoResearch) at 4 °C for 30 min. Cells were analysed on an Attune NxT Flow Cytometer (ThermoFisher Scientific) and analysed using FlowJo software.

### Peptide uptake assay
24 h post-transfection, HaCaT cells were plated on 10 cm dishes in full growth medium for 24 h, prior to treatment with 0.1 μg/ml FAM-conjugated CAR peptide (CAR-FAM), 0.1 μg/ml FAM-conjugated *m*CAR

peptide (*m*CAR-FAM) or vehicle control (Nil). Following 4 h incubation, cells were detached using enzyme-free Hanks'-based cell dissociation buffer (Invitrogen), to ensure retention of peptide associated with trypsin-cleavable cell surface HSPGs as well as internalised peptide. Cells were washed three times with 0.1/0.1 buffer, resuspended in PBS and analysed on an Attune NxT Flow Cytometer (ThermoFisher Scientific) and analysed using FlowJo software. To normalise data between replicate experiments, mean fluorescence intensity was normalised relative to the signal for *m*CAR-FAM in control knockdown cells.

## Scratch wound cell migration analysis

Glass-bottom 24-well plates were coated with plasma fibronectin from solution (10 μg/ml). HaCaT keratinocyte cells were plated at near confluence in fibronectin-coated wells for 24–36 h in DMEM supplemented with 1% FBS. Cell monolayers were wounded with sterile pipette tips and washed twice with medium, prior to treatment with 10 μg/ml CAR or *m*CAR peptides or vehicle control in DMEM supplemented with 1% FBS. Time-lapse brightfield images were acquired on a 3i Marianas live-cell imaging system using a Zeiss 10x/NA0.45 Plan-Apochromat objective or Zeiss Apotome2 widefield microscope using a Zeiss 10x/NA0.3 M27 EC Plan-Neofluar objective. Point visiting was used to allow multiple positions to be imaged within the same time-course and cells were maintained at 37 °C and 5% CO$_2$. Images were collected every 10 min for up to 21 h using a Hamamatsu ORCA-Flash4.0 v2 sCMOS camera on the 3i Marianas or a Zeiss Axiocam 506 mono on the Zeiss Apotome2. Percentage scratch wound closure was calculated by using ImageJ to measure the area of the gap between leading edges of each cell monolayer at the beginning and end of the time-lapse movie. Individual cell migration was tracked manually using the MTrackJ plugin for ImageJ. Migration tracks were plotted using the Chemotaxis Tool and Manual Tracking ImageJ plugins.

For experiments investigating the effect of CYTH2 knockdown on HaCaT keratinocyte migration, cells were seeded at near confluence in 96 well plates coated with fibronectin, 24 h after transfection. 36 h post-transfection, cells were serum-starved for 4 h and monolayers wounded with sterile pipette tips and washed twice with medium, prior to treatment with 10 μg/ml CAR or mCAR peptides or vehicle control in DMEM. Cells were imaged in triplicate wells in 96-well plates using an Incucyte® S3 Live-Cell Analysis System (Sartorius) with a 10x objective once every hour. Migration of the wound edge was analysed using imageJ.

## siRNA-mediated knockdown transfections

siRNA-mediated knockdown was achieved by using an Amaxa Nucleofector IIb system according to manufacturer's instructions. Briefly, TIF or HaCaT cells were cultured to 70% density before trypsinisation and approximately $2.5 \times 10^6$ cells were used per reaction. TIF cells were transfected using Nucleofector Kit NHDF, programme A-023 and 300 nM oligonucleotides. HaCaT cells were transfected using Nucleofector Kit V, programme U-020 and 300 nM or 600 nM oligonucleotides. Cells were transfected with 300 nM human *SDC4* siRNA (Ambion Silencer Select Oligo #1 s12638 or Oligo #2 s12639), 300 nM human *ARF6* siRNA (Dharmacon, sequence (sense) 5′- CGGCAUUA-CUACACUGGGA-3′), 600 nM human *CYTH2* siRNA (Ambion Silencer Select Oligo s225107) or either 300 nM or 600 nM AllStars Negative Control siRNA (Qiagen). Cells were subject to two rounds of knockdown transfection with 48-h intervals. Experiments were performed 48 h after the second transfection. Levels of protein knockdown were assessed by flow cytometry or immunoblotting.

## ARF6 effector pull down assay

Active ARF6 was assessed as described previously[11,13]. GST-GGA3 was isolated from JM109 E. coli (Stratagene) following IPTG-induced protein expression. Soluble bacterial lysate was incubated with pre-washed Glutathione Sepharose beads (GE Healthcare) for 1 h at RT, to allow GST-GGA3 protein binding. GST-GGA3-bound Sepharose beads were stored at −80 °C. To test the role of CAR stimulation on ARF6 activity, HaCaT cells were plated in DMEM containing 10% FBS for at least 16 h prior to 4 h serum starvation and subsequent stimulation with CAR peptide, *m*CAR peptide or vehicle control for 0, 10, 30, 60 or 120 min before lysis. A reverse time-course of stimulation was performed to enable simultaneous lysis of all cell samples. Cells were lysed in either 50 mM Tris-HCl pH7.2, 150 mM NaCl, 10 mM MgCl$_2$, 1% Nonidet P-40 substitute, 2.5% glycerol or 50 mM Tris pH = 7.5, 150 mM NaCl, 10 mM MgCL$_2$, 1% Triton X-100, 0.5% Deoxycholate, 0.1% SDS, 10% glycerol. Lysis and wash buffers were supplemented with phosphatase inhibitors (50 mM NaF, 10 mM NaVO$_4$) and protease inhibitors (0.5 mM AEBSF, 2 μg/mL leupeptin, 2 μg/mL aprotinin and cOmplete™ EDTA-free protease cocktail, Sigma Aldrich). Clarified lysates were incubated for 60 min with GST-GGA3-bound Sepharose beads at 4 °C. Beads were washed 3 times in wash buffer (50 mM Tris pH = 7.5, 100 mM NaCl, 2 mM MgCL$_2$, 1% IGEPAL CA-630, 10% glycerol)[71] and eluted with SDS sample buffer. Levels of bound active ARF6 were detected by immunoblotting. Total cell lysates were used to determine total ARF6 and tubulin levels.

ARF6 was detected using 1 μg/ml mouse anti-ARF6 monoclonal antibody (ARFAG; #A5230, Merck) and tubulin was detected with 1 μg/ml mouse anti-Tubulin monoclonal antibody (DM1A; #T9026, Merck). Primary antibodies were fluorescent-detected with AlexaFluor-680-conjugated Goat anti-Mouse IgG (H + L) highly cross-adsorbed secondary antibody (1/5000, 0.4 μg/ml; #A21058, Invitrogen). Proteins were detected and quantified using the Odyssey western blotting fluorescence detection system (LI-COR Biosciences), as described previously[12]. Uncropped and unprocessed scans of the blots are available in the Source Data file.

## Single-cell RNA-Seq

An extensive scRNA-Seq analysis of cutaneous wound healing in mouse by Haensel et al. was taken for analysis[23]. The experiments include scRNA-Seq data of three wounded mouse skin samples, with a total of 16,428 cells and 27,998 genes catalogued, which were taken for further analyses. Subsequent analyses were performed with the SCANPY Python library[72], where default settings were used unless otherwise stated. Cells were filtered based on read counts (800 < x < 45,000) and number of genes expressed (x > 300), and genes were filtered based on number of cells expressed in (x > 0.5%), resulting in a filtered set of 16,351 cells and 13,807 genes. Counts were log-transformed (SCANPY function 'pp.log1p'), batch correction was performed using the Python implementation (Pedersen 2012) of the ComBat algorithm[73] ('pp.combat') based on the three samples as independent batches. Highly variable genes (HVGs) were determined ('pl.highly_variabl_genes') using the 'seurat_v3' flavour and the 'counts' layer, with the top 6000 HVGs kept for subsequent analyses. Principal component analysis ('pp.pca') was performed using the HVGs and 40 components, and nearest neighbours ('pp.neighbours') calculated with local neighbourhood size of 15. The Python implementation (Traag 2017) of Louvain clustering[74] was used to cluster cells at $r = 0.75$ resolution. Uniform manifold approximation and projection (UMAP; 'tl.umap') modelling was used to visualise the clusters in 2D space. Ranked lists of genes and corresponding log-fold changes were generated for all cell clusters identified by Louvain clustering ('tl.rank_genes_groups'), which were subsequently used to assign cell type annotations based on review of existing literature. Cells which were identified as satellite cells or skeletal muscle were removed from the final analysis and figures.

## Co-Immunoprecipitation for proteomic analysis

For proteomic analysis, approximately $5 \times 10^7$ cells for each cell line (Syn4WT, Syn4-/-, Syn4Y180E and Syn4Y180L MEFs) were lysed in

800 ml of 0.5% Igepal buffer (0.5% Igepal CA-630, 10 mM Tris-HCl pH 7.5, 250 mM NaCl, 0.5 mM EGTA, 10% glycerol, 10 mM sodium fluoride, 5 mM sodium orthovanadate, 10 µg/ml leupeptin, 10 µg/ml aprotinin and 0.5 mM AEBSF). Clarified lysates were incubated 3µg mouse anti-huSDC4 antibody (5G9; sc-12766, Santa Cruz) with 80 ml Protein G sepharose beads for 1 h at 4 °C. Beads were washed 3 times with lysis buffer, eluted with 80 ml 5x reducing sample buffer and incubation at 95 °C for 5 min. Immune-complexes were separated by SDS/PAGE and stained with Instant Blue (Expedeon). Each sample was fractionated into 20 slices and de-stained with repeated incubations of 50% (v/v) Acetonitrile (ACN)/ 50% 25 mM NH4HCO3 (v/v) at RT. Gel pieces were dehydrated by 2 × 5 min incubations with ACN followed by vacuum centrifugation. Peptides were reduced via incubation with 10 mM dithiothreitol (DTT) at 56 °C for 1 h, followed by alkylation by incubating with 55 mM iodoacetamide (IA) for 45 min at RT while protected from light. DTT and IA were removed by two rounds of washing and dehydration: 5 min incubations with 25 mM NH4HCO3 followed by a 5 min incubation with ACN. ACN was removed by centrifugation and gel pieces dried via vacuum centrifugation. 1.25 ng/L porcine trypsin (Promega, Cat No:V5280) in 25 mM $NH_4HCO_3$ was added to the gel pieces, which were incubated at 4 °C for 45 min followed by 37 °C overnight. Trypsinised peptides were collected from gel pieces via centrifugation, followed by further extraction 30 min incubation at RT with 99.8% (v/v) ACN/0.2% (v/v) formic acid and then 50% (v/v) ACN/ 0.1% formic acid (v/v) followed by centrifugation. Collected eluates were dried by vacuum centrifugation and re-suspended in 10 µl 5% (v/v) ACN in 0.1% formic acid per fraction.

Peptide analysis by LC-MS/MS was performed using an UltiMate 3000 Rapid Separation LC (Dionex Corporation) coupled to an Orbitrap Elite mass spectrometer (Thermo Fisher). Peptides were selected for fragmentation automatically by data-dependent analysis. Data produced were searched using Mascot (Matrix Science UK), against the uniprot.2011-05-03 mammalia database, with fragment ion mass tolerance of 0.50 Da and parent ion tolerance of 10.0 PPM.

Scaffold 4 (Proteome Software) was used to validate MS/MS-based peptide and protein identifications. Peptide identifications were accepted at >95.0% probability by the Peptide Prophet and protein identifications were accepted at >99.0%. Proteins that contained similar peptides and could not be differentiated based on MS/MS analysis alone were grouped to satisfy principles of parsimony. The mass spectrometry proteomics data have been deposited to the ProteomeXchange Consortium via the PRIDE partner repository with the dataset identifier PXD046898 and doi: 10.6019/PXD046898.

### Co-Immunoprecipitation for immunoblot analysis

A total of $1 \times 10^7$ HaCaT cells per condition were lysed in 2 ml of 1% Igepal buffer (1% Igepal CA-630, 25 mM Tris-HCl pH 7.4, 150 mM NaCl, 1 mM EDTA, 10% glycerol, 10 µg/ml leupeptin, 10 µg/ml aprotinin and 0.5 mM AEBSF). 100 ml of total cell lysate was retained for immunoblotting, while 1900 ml clarified lysates were incubated 10 µg rabbit anti-ARF6 monoclonal antibody (D12G6; #5740, Cell Signalling Technologies) with 30 ml Protein A sepharose beads for 2 h at 4 °C. Beads were washed twice with lysis buffer and once with cold PBS, prior to elution with 30 ml 2x reducing sample buffer and incubation at 100 °C for 5 min. Immune-complexes and total cell lysates were separated by SDS/PAGE and subjected to immunoblotting. Membranes were cut at approximately 30 kDa and 60 kDa to enable identification of ARF6 (approximate molecular weight: 18 kDa) and CYTH2 (approximate molecular weight: 46 kDa) in the same immunoprecipitation samples, using 1 µg/ml mouse anti-ARF6 (ARFAG; #A5230, Merck) and 1 µg/ml mouse anti-CYTH2 (6H5; #H00009266-M02, Abnova) antibodies, respectively. ARF6 was detected in total cell lysates using 1 µg/ml rabbit anti-ARF6 (D12G6; #5740, Cell Signalling Technologies). Primary antibodies were detected using HRP-linked goat anti-mouse (0.2 µg/ml,

1:5000; #31160, Pierce) or HRP-linked goat anti-rabbit (0.2 µg/ml, 1:5000; #31210, Pierce) secondary antibodies, followed by enhanced chemiluminescence (ECL) using SuperSignal™ West Femto Maximum Sensitivity Substrate (#34094, Thermo Scientific) imaged on a Bio-Rad Chemidoc Touch Imaging System. Actin loading control was probed on a separate membrane due to actin and CYTH2 having similar molecular weights and the same species antibody. Actin was detected with 0.2 µg/ml mouse anti-actin (AC-40; #A3853, Merck) followed by AlexaFluor-790-conjugated Goat anti-Mouse IgG (H + L) highly cross-adsorbed secondary antibody (1/10,000, 0.2 µg/ml; A11357, Invitrogen). Protein levels were quantified using ImageJ or the Odyssey western blotting fluorescence detection system (LI-COR Biosciences). Uncropped and unprocessed scans of the blots are available in the Source Data file.

### Statistical analysis

The distribution of all in vivo data was determined by various normality tests (Shapiro-Wilk, Anderson-Darling, robust Jaque-Bera). Homogeneity of variances across groups was checked by Levene's and Bartlett tests. As our data deviated from Gaussian normal distribution, non-parametric tests were applied.

For comparison of two groups, Wilcoxon-Mann-Whitney test (with tie correction) was used. For comparison of multiple (>2) groups, Kruskal-Wallis rank sum test with post-hoc Dunn's test for pairwise comparisons of independent samples was used. For categorical data, Pearson´s chi-square test with post-hoc Fisher exact test was employed. The $p$-values were adjusted for multiple comparisons by the Holm method.

For proportion and percentage data, the groups were compared by Pearson's Chi-square test following a Fisher post-hoc test with holm-adjusted $P$-values. $P$-values and N-numbers for in vivo experiments are reported in figure legends.

For in vitro analyses, one-way or two-way ANOVA were performed using Tukey's method to correct for multiple comparisons. Where noted, $t$ tests were performed using the Holm-Sidak method. Statistical tests and $P$-values for in vitro experiments are reported in figure legends.

$P$-values < 0.05 were considered statistically significant. The significance level refers to two-tailed tests. In the figures the following scheme is used to express $p$-values: *denotes $P \le 0.05$; **$P \le 0.01$; ***$P \le 0.001$; ****$P \le 0.0001$. The statistical software R Bioconductor or GraphPad PRISM were used for conducting statistical tests.

### Reporting summary

Further information on research design is available in the Nature Portfolio Reporting Summary linked to this article.

## Data availability

The mass spectrometry proteomics data have been deposited to the ProteomeXchange Consortium via the PRIDE partner repository with the dataset identifier PXD046898. The scRNA-Seq data from cutaneous wound healing in mice[23] are available on the GEO database under accession code GEO: GSE142471. All other data supporting the findings this study are available within the paper, Supplementary Information and Source Data files. Further information and requests for resources and reagents should be directed to the Lead Contacts, Tero Järvinen (tero.jarvinen@tuni.fi) and Mark Morgan (mark.morgan@liverpool.ac.uk). Source data are provided with this paper.

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

## Acknowledgements

We thank Dr Venkata Ramana Kotamraju for peptide synthesis, Marianne Karlsberg, Anni Laitinen, Robbin Newlin, Fernanda Munoz Caro and Marja-Leena Koskinen for practical support and for performing the histochemical work. Guillermina Garcia is thanked for her technical expertise and help with quantitative microscopy. Dr Mark Bass (University of Sheffield, UK) provided SDC4 knockout mice (originally generated in the Centre for Animal Resources and Development, Kumamoto University, Japan by professor Tetsuhito Kojima, Nagoya University, Japan)[67]. Prof. Hiroyuki Sakagami (Kitasato University, Japan) provided anti-cytohesin-2 antibody. We thank the Biomedical Imaging Facility and Centre for Cell Imaging for support and access to equipment, provided by Liverpool Shared Research Facilities, University of Liverpool, and Dr Stacey Warwood, Dr Julian Selley and Dr David Knight, from the Biological Mass Spectrometry Core Facility at the University of Manchester (RRID code: SCR_020987), for running samples and data archiving/ management. The schematic diagram in Fig. 10 was created using elements from BioRender.com (wound image, vasculature and syringe). This work was funded by the US Armed Forces Institute for Regenerative Medicine (AFIRM) (E.R.), Sigrid Juselius Foundation (T.A.H.J.), the Academy of Finland (T.A.H.J.), Päivikki and Sakari Sohlberg Foundation (T.A.H.J.), Instrumentarium Research Foundation (T.A.H.J.), Tampere Tuberculosis Foundation (T.A.H.J.), Finnish Medical Foundation (T.A.H.J.), Pirkanmaa Hospital District Research Foundation (T.A.H.J.), the Finnish Cultural Foundation (T.A.H.J.), Breast Cancer Now (Grant Reference 2015MayPR507 - M.R.M. & H.M.), North West Cancer Research (Grant References CD2019.12 - M.R.M. & H.M., CR1010 - M.R.M. & K.I.W., CR1143 -M.R.M. & K.I.W., and CEMorgan2018 - M.R.M.) and Wellcome (Grant References 215191/Z/19/Z – M.R.M. & B.D.S., 092015 – M.J.H. & M.R.M.).

## Author contributions

Conceptualisation: S.P., M.J.H., E.R., M.R.M., T.A.H.J. Methodology: H.M., M..R.M., T.A.H.J. Investigation: H.M., B.D.S., H.R.B., U.M., M.V., R.K.B., K.I.W., C.M.L.T., T.P., T.K., M.R.M., T.A.H.J. Visualisation: H.M., U.M., H.R.B., M.R.M. Funding acquisition: M.J.H., E.R., M.R.M., T,A,H.J. Project administration: E.R., M.R.M., T.A.H.J. Supervision: E.R., M.R.M., T.A.H.J. Writing – original draught: H.M., B.D.S., U.M., E.R., M.R.M., T.A.H.J. Writing – review & editing: All authors.

## Competing interests

E.R. has ownership interest (including patents) in Vascular Biosciences Inc., biotech company developing the CAR peptide for clinical applications. No conflicts of interest were disclosed by the other authors.
