## [Peer Review File · Nature Communications]

REVIEWER COMMENTS

Reviewer #1 (Remarks to the Author):

This manuscript describes the acceleration of excision wound healing in mice by intravenous administration of CARSKNKDC peptide through stimulation epithelial cell migration. Data suggests that CAR induces SDC4-dependent activation of the small GTPase Arf6 (ARF6) and promotes SDC4- and ARF6-mediated keratinocyte migration. Genetic ablation of SDC4 in mice eliminates CAR-induced wound re-epithelialisation following systemic administration. The authors propose that CAR peptide activates SDC4 function to selectively promote re-epithelialisation.

Overall the information provided is important and interesting and is an extension of considerable work with CAR wound-homing peptides. The data supports the conclusions proposed by the authors. The methodology builds on standard animal wound healing models but applied advanced technology and methodology which is well described in the methods section of the submission. The presentation is free of spelling diction or other technical errors.

Concerns with the manuscript:

1. In line 119 the authors contend that wound contraction from the panniculosis carnosus could not contribute to the contraction measured but it is difficult to understand the basis for their conclusions.
2. In figure 5 J no error bars or stats are illustrated in the figure.
3. Technically twice per day tail vein injections for more than 10 days could be challenging in mice. How do the authors ensure that the drugs are successfully administered?
4. Were the histologic assessments performed in a blinded fashion and if so, can the authors indicate this in their submission.
5. Can the authors explain the use of Gamma interferon in the cell culture media used in the embryonic fibroblasts experiments where it is known to be anti proliferative?
6. Certain types of wounds heal largely by re-epithelialization whereas others require dermal contraction and remodelling for strength of closure. Do the authors have data that allows them to propose this epithelial based stimulation of healing to support the broad contention that all wound healing is enhanced with CAR therapies such as other wound models including incisions with breaking strength assessments ect?
7. Data describing cytoskeletal reorganization after CAR stimulations and the specificity of the effects and $\alpha 5 \beta 1$ integrin adhesion complex would be worthy of inclusion the manuscript.

Reviewer #2 (Remarks to the Author):

In this manuscript, the authors describe the promotion of wound healing by CAR peptides and show that SDC4 is required for this process. On the other hand, wound healing involves cell motility and integrin recycling, and ARF6 is a fundamental G-protein that regulates these events. It has been described in the literature, a long ago, that cell-ECM adhesion and SDC4 are closely related to ARF6 activation. However, it is not yet clear what pathways ECM adhesion and SDC4 use to activate ARF6, or which ARF6GEF(s) activates ARF6 by SDC4 or by ECM adhesion. This reviewer feels that identification of the molecular mechanism by which CAR/SDC4 activates ARF6, including the identification of the ARF6GEF(s), should be the core part of this manuscript.

Reviewer #3 (Remarks to the Author):

This manuscript by Savage et al. describes the discovery of a novel wound-homing peptide CAR (CARSKNKDC). It is shown that the CAR peptide (facilitates or encourages) wound healing by promoting keratinocyte migration in vitro, and mediates cytoskeletal remodeling. This is accomplished by CARs SDC4-dependent activation of the small GTPase Arf6 (ARF6). With that being stated, Savage et al. propose that the CAR peptide may provide an entirely new therapeutic approach to enhance wound healing.

The title and abstract are appropriate for the content of the text. Additionally, the article is well constructed, the experiments were well conducted, and the analysis was well performed. The reported results represent a notable advance in the development of wound-homing peptides and their mechanism of action. In my opinion, the manuscript is suitable for publication in Nature Communication after the authors have addressed the following comments and questions:

- 1.The introduction should be expanded with a discussion of the scope and significance of the issue and or problem, including a brief review of the literature that should provide the reader with a synthesis of previous work.
- 2.It has been shown that CAR peptide-treated wounds have substantially fewer myofibroblast than controls and other treatment groups. The reason for this observation should be explained.

3. The wound mouse model needs a more detailed explanation. Is the mechanism of wound healing by the CAR peptide dependent on the size and position of the wound?

4. The relevance and point behind extensive single-cell RNA-Seq (scRNA-Seq) analysis of cutaneous wound healing in mouse studies that were performed are not clear in their reasoning

5. The experiment shown in Figure 5, immunofluorescence micrographs demonstrating that internalized CAR co-localizes with SDC4, would benefit from including a mCAR-FAM control.

6. Figure 2. shows that the CAR peptide stimulates wound re-epithelialization. It is interesting to note that the CAR peptide has an effect in reducing open wound gaps, as soon as 5 days post wounding (figure 2B), while the wound width is not affected until day 10 (Figure 2E). How to interpret this and what is the reason for this difference needs to be addressed.

Reviewer #1:

This manuscript describes the acceleration of excision wound healing in mice by intravenous administration of CARSKNKDC peptide through stimulation epithelial cell migration. Data suggests that CAR induces SDC4-dependent activation of the small GTPase Arf6 (ARF6) and promotes SDC4- and ARF6-mediated keratinocyte migration. Genetic ablation of SDC4 in mice eliminates CAR-induced wound re-epithelialisation following systemic administration. The authors propose that CAR peptide activates SDC4 function to selectively promote re-epithelialisation.

Overall, the information provided is important and interesting and is an extension of considerable work with CAR wound-homing peptides. The data supports the conclusions proposed by the authors. The methodology builds on standard animal wound healing models but applied advanced technology and methodology which is well described in the methods section of the submission. The presentation is free of spelling diction or other technical errors.

Concerns with the manuscript:

Comment 1. In line 119 the authors contend that wound contraction from the panniculosis carnosus could not contribute to the contraction measured but it is difficult to understand the basis for their conclusions.

Response: We acknowledged the general role of panniculus carnosus-driven wound contraction in the manuscript (p. 6, second paragraph and p. 27, second paragraph). However, several observations led us to reach the conclusion that wound contraction overall was not significantly affected by CAR treatment:

1) Overall wound width (W_x) as a direct measure of wound contraction was not reduced by CAR treatment at early timepoints (days 5 and 7) (New Fig 2E). Whereas CAR-induced substantial changes in other metrics associated with overall wound closure/re-epithelialisation at these early timepoints: Specifically, open wound gap, (WG_x ; Fig 2B), complete re-epithelialisation (New Fig 2F) and, most importantly, hyperproliferative

epidermis length (E1x+E1X, New Fig2C) and area (New Fig2D). Thus, re-epithelialisation was significantly accelerated by CAR peptide.

2) The number of myofibroblasts, the cell type number was not affected by CAR treatment (Supplementary Fig S1; p. 9, first paragraph). To eliminate the possibility that CAR peptide might promote wound repair by initiating early myofibroblast activation, in the revised manuscript we now include α -SMA IHC images from day-7 wounds demonstrating the absence of myofibroblasts in granulation tissue in all treatment conditions (New Supplementary Fig S3).

3) We show that the CAR target, SDC4 is only expressed in the epidermis and blood vessels of wounds: "IHC double staining for SDC4 and fibronectin showed that, where SDC4 was localised within migrating epidermis, the expression pattern of fibronectin is the opposite; there is no expression in migrating epidermis but abundant expression throughout underlying granulation tissue (New Fig. 5A,C-E & G-I)" (p. 12, first full paragraph).

We feel that these results amply justify our conclusion that wound contraction is not a factor in the biological response to CAR peptide. While panniculus carnosus- and myofibroblast-driven contraction, can contribute to wound healing, none of our data suggest they contribute to CAR-dependent increased wound repair.

Comment 2. In figure 5 J no error bars or stats are illustrated in the figure.

Response: We thank the reviewer for pointing this out. In fact, the original figure did include error bars, however, they were so small for several conditions that the lines did not render properly when the graphs were copied from GraphPad Prism into Illustrator, meaning the lower error bars were obscured by the data bars. We have now rectified this issue and increased the line thickness (Revised Fig. 5J).

Comment 3. Technically twice per day tail vein injections for more than 10 days could be challenging in mice. How do the authors ensure that the drugs are successfully administered?

Response: The reviewer is right. The injection regimen was technically demanding. However, the following measures were taken to ensure successful delivery of the drugs:

- 1) All tail vein injections were performed by a single scientist who had years of daily experience with tail vein-injections.
- 2) BALB/c mice were used for the long treatment trials because the tail veins are easier to see, than in other strains (e.g. C57BL/6 mice).
- 3) All injections were carried out under anaesthesia and the tails were warmed up in a water bath to facilitate proper blood circulation and tail vein dilation.
- 4) Stasis and vein immobilization were obtained by placing a finger proximal to the injection site.
- 5) The left tail vein was used in the morning and the right vein in the evening.
- 6) The injections were started from the tip of the tail and moved along proximally.
- 7) Only one injection was carried out at a time. In the rare cases where extravasation was identified, the injection was stopped, and the rest of the injection was administered s.c. as far as possible from the wounds to obtain systemic administration.

We have now described these precautions in more detail in the Methods section (p.28).

Comment 4. Were the histologic assessments performed in a blinded fashion and if so, can the authors indicate this in their submission.

Response: Yes, the studies were conducted in a blinded fashion, when appropriate. We state in the manuscript: *“All analyses were carried out by adhering to ARRIVE 2.0 guidelines⁶³.”* (p. 24). We have now added to the text that the guidelines include the blinded analyses (Methods, p. 31, end of first paragraph).

Comment 5. Can the authors explain the use of Gamma interferon in the cell culture media used in the embryonic fibroblasts experiments where it is known to be anti-proliferative?

Response: Interferon gamma (IFN γ) was used for culturing the mouse embryonic fibroblast (MEF) cell lines, because they are derived from the SDC4 $^{-/-}$ immortomouse model, which enables interferon-inducible expression of a thermolabile large tumor antigen (TAg). Thus, during routine cell culture, the MEFs were maintained in the presence of IFN γ and

at the permissive temperature of 33°C. However, all assays were performed in the absence of IFN γ , to ensure it did not impact the behaviour of the cells.

These culture and experimental conditions are the same as earlier studies using these cells and their derivatives (Refs 13 and 20). Importantly, as all of the MEF cell lines were cultured in the presence of IFN γ , it is unlikely to have affected the results of downstream experiments.

Comment 6. Certain types of wounds heal largely by re-epithelialization whereas others require dermal contraction and remodelling for strength of closure. Do the authors have data that allows them to propose this epithelial based stimulation of healing to support the broad contention that all wound healing is enhanced with CAR therapies such as other wound models including incisions with breaking strength assessments ect?

Response: The excision wound model is the most-commonly used model in skin wound studies. BALB/c mice were selected for the study because re-epithelialisation contributes more to wound closure than panniculus carnosus-driven wound contraction (approximately 60% and 40% contribution, respectively) in BALB/c mice than in other common mouse strains. Also, the tail veins are easier to see than in other strains. Furthermore, the wound healing experiments on the SDC4 WT vs KO mice were carried out in C57BL/6 mice. We demonstrate also in this strain that CAR peptide accelerates wound closure and re-epithelialisation. Thus, our selected wound model is the “golden standard” experimental wound healing model in rodents and we reproduced the data in the two most commonly used mouse strains. However, we have not had the opportunity to investigate the impact of CAR treatment in other skin wound or other injury models. This is something that we would be keen to pursue in the future, but is not within the scope of this existing study.

We have now given the rationale for the selection of this model in the Methods (p. 7, first paragraph, p. 28, first and middle paragraphs). We also highlight the need to investigate the effect of CAR in alternative models of wound and tissue repair in the Discussion (p. 26, last paragraph).

Comment 7. Data describing cytoskeletal reorganization after CAR stimulations and the specificity of the effects and $\alpha 5\beta 1$ integrin adhesion complex would be worthy of inclusion in the manuscript.

Response: In the original version of the manuscript, imaging data on CAR-induced cytoskeletal reorganization and $\alpha 5\beta 1$ integrin in MEFs expressing endogenous SDC4 ($Im^{+/+}$), SDC4^{-/-} MEFs and SDC4WT re-expressing MEFs were presented on pages 10-11 and included several supplementary figures (Old Supplementary Figs S3A/B & S5A/B; now New Supplementary Figs S4A/B & S6A/B). This included specificity controls include the *mCAR* peptide, which is an inactive variant of CAR, as well as siRNA and knockout studies. These cells were used as they provide a good model for examining SDC4-dependent functions.

However, as our data suggested that CAR selectively targets wound epithelia, we decided not to do perform further quantitative analyses on MEFs. Instead, we now include new quantitative data, demonstrating that CAR peptide triggers a rapid loss of $\alpha 5\beta 1$ integrin from well-established adhesion complexes (New Fig. 4C/D). However, 30 minutes of sustained CAR treatment, resulted in redelivery of $\alpha 5\beta 1$ integrin to the cell-matrix interface in robust adhesion complexes (New Fig. 4C/D). Thus, CAR peptide regulates $\alpha 5\beta 1$ integrin adhesion complex dynamics in epithelial cells (New Fig. 4C/D) and promotes ARF6- and SDC4-dependent epithelial cell migration (Fig. 7).

Intriguingly, the temporal profile of $\alpha 5\beta 1$ integrin redistribution, mirrors the initial inactivation of ARF6 followed by a wave of ARF6 activation induced by CAR treatment. This would be consistent with the role of ARF6 in regulating $\alpha 5\beta 1$ recycling that we reported previously (Ref. 13).

Reviewer #2:

In this manuscript, the authors describe the promotion of wound healing by CAR peptides and show that SDC4 is required for this process. On the other hand, wound healing involves cell motility and integrin recycling, and ARF6 is a fundamental G-protein that regulates these events. It has been described in the literature, a long ago, that cell-ECM adhesion and SDC4 are closely related to ARF6 activation. However, it is not yet clear what pathways ECM adhesion and SDC4 use to activate ARF6, or which ARF6GEF(s) activates ARF6 by SDC4 or by ECM adhesion. This reviewer feels that identification of the molecular mechanism by which CAR/SDC4 activates ARF6, including the identification of the ARF6GEF(s), should be the core part of this manuscript.

Response: Generally speaking, we consider the key message of the paper to be that the CAR peptide may make it possible to enhance wound healing through *systemic* treatment. Current efforts to enhance tissue repair in the skin and other tissues almost exclusively rely on local application of therapeutics. The CAR peptide was the first introduced to the fields of surgery and traumatology in hopes of using it to deliver systemically administered therapeutics to sites of injury. Reporting the unexpected finding that CAR possesses inherent regenerative activity in wound healing; enhancing wound re-epithelialization and accelerating wound healing. Thus, we would argue that the translational potential of our finding should be the core of the manuscript.

However, the reviewer's point is an important one, so we have exerted significant efforts to gain further mechanistic insight into how CAR peptide and SDC4 modulate ARF6 activity. Together, our new data demonstrate that CAR promotes its pro-migratory activity via the guanine nucleotide exchange factor (GEF) CYTH2 (also known as cytohesin-2 or ARNO). These data are presented in New Fig 8A-K and New Supplementary Fig S7 A-E and described on pages 16-18. In summary, we now show that CAR peptide regulates the association of CYTH2 with ARF6, and that CYTH2 is required for CAR-dependent ARF6 activation and epithelial cell migration.

We initially employed a preliminary proteomic approach to identify potential ARF regulatory molecules co-immunoprecipitating with SDC4, or mutants of SDC4 known to differentially

modulate ARF6 activity (Syn4Y180E and Syn4Y180L) (New Fig 8A/B). This initial screen allowed us to identify three potential candidate GEFs, which may be functionally associated with SDC4. Of these three GEFs (CYTH2, IQSEC1 and CYTH3), scRNA-seq data of mouse wound tissue revealed that only CYTH2 exhibited elevated expression in epithelia (which also exhibit enhanced SDC4 and ARF6 expression) (New Supplementary Fig. S7A-C).

Follow-up experiments, using quantitative imaging analysis and co-immunoprecipitation, demonstrated that stimulation with CAR peptide promotes the co-localisation and co-association of CYTH2 with ARF6 (New Fig 8C-E). Moreover, siRNA-mediated CYTH2 knockdown inhibited CAR-dependent ARF6 activation (New Fig 8F/G) and CAR-stimulated HaCaT migration (New Fig 8H-K). Thus, we now postulate that CAR peptide promotes epithelial migration and wound healing by modulating SDC4- and CYTH2-dependent ARF6 activity.

Intriguingly, we also found that CAR treatment reduced co-localisation of IQSEC1 and ARF6 (New Supplementary Fig S7D-E). Suggesting that CAR may trigger a switch from IQSEC- to CYTH2-mediated ARF6 activity modulation. Obviously, these data raise lots new and important questions about how the network of GEFs and GAPs may co-ordinate SDC4-dependent functions spatially and temporally. We consider these questions beyond the scope of the current manuscript, but are keen to explore them in the future. So we would like to thank the reviewer for their insightful comments.

Reviewer #3:

This manuscript by Savage et al. describes the discovery of a novel wound-homing peptide CAR (CARSKNKDC). It is shown that the CAR peptide (facilitates or encourages) wound healing by promoting keratinocyte migration in vitro, and mediates cytoskeletal remodeling. This is accomplished by CARs SDC4-dependent activation of the small GTPase Arf6 (ARF6). With that being stated, Savage et al. propose that the CAR peptide may provide an entirely new therapeutic approach to enhance wound healing.

The title and abstract are appropriate for the content of the text. Additionally, the article is well constructed, the experiments were well conducted, and the analysis was well performed. The reported results represent a notable advance in the development of wound-homing peptides and their mechanism of action. In my opinion, the manuscript is suitable for publication in Nature Communication after the authors have addressed the following comments and questions:

Comment 1. The introduction should be expanded with a discussion of the scope and significance of the issue and or problem, including a brief review of the literature that should provide the reader with a synthesis of previous work.

Response: Our study touches several fields, including wound healing, phage library screening for homing peptides, targeted drug delivery, heparan sulfate proteoglycans, small GTPases, cell migration and treatment of tissue injuries. So it was difficult to choose which areas to cover in detail. Instead, we chose to describe the path and therapeutic rationale that led us to the current study, and its purpose. We have then dealt with other relevant work and information when the topic came up in the Results and the Discussion.

Comment 2. It has been shown that CAR peptide-treated wounds have substantially fewer myofibroblast than controls and other treatment groups. The reason for this observation should be explained.

Response: Myfibroblasts were quantified to rule out the possibility that the accelerated wound closure by CAR peptide was caused by myofibroblast-driven wound contraction. Surprisingly, myofibroblast number was reduced in CAR peptide treated wounds. The mechanism for this is not known, but for the purposes of the current study, it helps rule out myofibroblast driven wound contraction as a basis of the CAR effect on wound closure.

Moreover, new α -SMA IHC data from day-7 wounds demonstrate that there are no myofibroblasts in granulation tissue in all treatment conditions (New Supplementary Fig S3). Eliminating the possibility that CAR peptide might initiate early myofibroblast activation, which is then suppressed by day-10.

The fact that CAR treatment suppresses myofibroblast activation, also suggests that CAR may inhibit excessive scarring. These points are discussed in the Results section (p. 9, first paragraph) and Discussion (p. 23).

Comment 3. The wound mouse model needs a more detailed explanation. Is the mechanism of wound healing by the CAR peptide dependent on the size and position of the wound?

Response: We used standard excision (6 mm) wound model, which is the most common wound healing model. We have described the model in detail (p. 28) and we now provide additional information regarding both wounding and peptide administration (p. 28).

The enhanced wound healing by CAR peptide was demonstrated in two different mouse strains, BALB/c and C57BL/6. In addition to data presented in the manuscript, we have performed a preliminary treatment trial on 4 mm excision wounds. While not included in the manuscript, the study recapitulated the enhanced wound closure by CAR treatment. For the purpose of this study, we focused on thorough demonstration of the CAR effect on wound healing in the 6mm excision wound model and identification of the target molecule of CAR in wounds. As mentioned by the reviewer, there are many details about the CAR activity that could not be covered in this first paper on the topic.

Comment 4. The relevance and point behind extensive single-cell RNA-Seq (scRNA-Seq)

analysis of cutaneous wound healing in mouse studies that were performed are not clear in their reasoning.

Response: Our *in vivo* results indicated that the biological activity of CAR was cell type-specific: Re-epithelialisation by keratinocytes was enhanced, but scar formation by fibroblasts was not. Immunohistochemistry showed that the CAR target (receptor), SDC4, is expressed in wound keratinocytes and blood vessels, but not in granulation tissue fibroblasts. The scRNA-seq studies were conducted to confirm this cell-type specific expression pattern in wound. The rationale is given in the text (p. 12, last paragraph and p.13, first paragraph, p. 14 second paragraph, p. 17, third paragraph).

Comment 5. The experiment shown in Figure 5, immunofluorescence micrographs demonstrating that internalized CAR co-localizes with SDC4, would benefit from including a mCAR-FAM control.

Response: We did not include *mCAR-FAM* in the immunofluorescence analysis, because it does not bind to cells (Ref. 4). However, we did include *mCAR-FAM* in the flow cytometric analysis of peptide binding to HaCaT cells (Fig 5J). These experiments confirmed negligible levels of *mCAR* binding to epithelial cells and also that SDC4 expression was required CAR to bind to cells.

Comment 6. Figure 2 shows that the CAR peptide stimulates wound re-epithelialization. It is interesting to note that the CAR peptide has an effect in reducing open wound gaps, as soon as 5 days post wounding (figure 2B), while the wound width is not affected until day 10 (Figure 2E). How to interpret this and what is the reason for this difference needs to be addressed.

Response: We have now added data on the length of hyperproliferative epidermal tongues to help deal with this issue (New Fig 2C). It is important to note that the wound width (W_x) is primarily an indicator of overall wound contraction. Whereas the hyperproliferative epidermis (HPE) area and length ($E_{1x}+E_{2x}$) as well as the gap between the epidermal tongues provide an indication of the level of re-epithelialisation.

The hyperproliferative epidermal tongues ($E_{1x}+E_{2x}$) were longer in the CAR treatment group, than in the *m*CAR and Control groups, while the wound width (W_x) was the same between all treatment groups at days 5 and 7. Therefore, these data suggest CAR peptide significantly accelerated re-epithelialisation, but had little impact on wound contraction. We have also now updated the descriptions in the text, to clarify these points (p. 7-8).

REVIEWERS' COMMENTS

Reviewer #1 (Remarks to the Author):

The revised manuscript supplemental figures and responses to the reviewers have been reviewed. The response to questions posed is very robust and acceptable. No further questions remain.

Reviewer #2 (Remarks to the Author):

With this revision, the authors have adequately responded to this reviewer's comments and the paper may be published.

Reviewer #3 (Remarks to the Author):

The revised version of the manuscript adequately addresses the points